# Hidden Breakthroughs in Language Model Training

**Sara Kangaslahti**
Harvard University
sarakangaslahti@g.harvard.edu

**Elan Rosenfeld**
Google Research
elanr@google.com

**Naomi Saphra**
Harvard University
nsaphra@fas.harvard.edu

## Abstract

Loss curves are smooth during most of model training, so visible discontinuities stand out as possible conceptual breakthroughs. These breakthroughs enable a deeper understanding of the model's concept structure, but only when they are properly identified. This paper argues that similar breakthroughs occur frequently *throughout* training, but they are obscured by a loss metric that collapses all variation into a single scalar. To find these hidden transitions, we introduce POLCA, a method for decomposing changes in loss along arbitrary bases of the low-rank training subspace. We use our method to identify clusters of samples that share similar changes in loss during training, disaggregating the overall loss into that of smaller groups of conceptually similar data. We validate our method on synthetic arithmetic and English language modeling, showing that POLCA recovers clusters that represent interpretable breakthroughs in the model's capabilities. We demonstrate the promise of these hidden breakthroughs as a tool for unsupervised interpretability.

## 1 Introduction

As large language models train, various internal structures develop during abrupt breakthroughs. These sudden drops in loss reveal the formation of mechanisms for in-context learning (Olsson et al., 2022b), natural language grammar (Chen et al., 2024a), hierarchical generalization (Murty et al., 2023), and many other concepts (McGrath et al., 2022; Lovering et al., 2022; Power et al., 2022; Abbe et al., 2021). However, the loss curve as a whole remains stubbornly smooth. As a result, these momentary conceptual breakthroughs are treated as isolated curiosities, while the majority of training behavior is considered predictable.

These breakthroughs—often colloquially termed *phase transitions* (Olsson et al., 2022a; Chen et al., 2024a; Murty et al., 2023)—are extremely consequential for our understanding of neural networks. Phase transitions represent critical periods of learning, so they offer key insights for training and optimization. For instance, introducing noisy data or changing the optimizer during a phase transition can significantly reduce the downstream performance of a model (Achille et al., 2017; Chen et al., 2024a). Their timing is used as evidence the role of specific mechanisms in model capabilities (Zhong et al., 2023; Olsson et al., 2022b; Chen et al., 2024a). If we could find more of these momentary training events, it could expand our understanding even further.

Prior work identifies breakthroughs through a *top-down* approach by measuring the training dynamics of a predefined concept or skill and searching for sudden changes. We instead propose a *bottom-up* unsupervised method for finding breakthroughs by grouping data points that have similar training behavior. This data-centric approach can be used to inform optimization choices such as data selection or learning rate scheduling. Like other bottom-up interpretability methods such as SAEs, PCA, and transcoders, our method seeks concepts that are used naturally by the model, rather than imposing an assumed structure onto learning and representation.

This work shows that in fact, a model undergoes many breakthroughs during training, but most are concealed when averaging all data into a single loss curve. Instead of averaging, we divide up the loss curve in two different ways to find hidden breakthroughs. First, we **disaggregate** the **aggregate** loss into losses on individual examples. By clustering the individual loss curves, we identify subsets of data that experience synchronized changes in loss, implying that they rely on the same conceptual breakthrough. However, any individual example might benefit from *multiple*

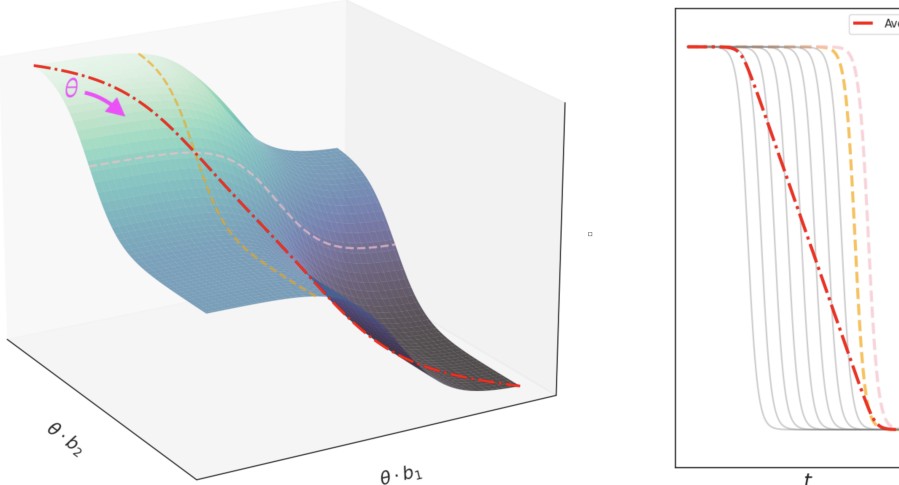

Figure 1: A smooth loss function may change sharply for a particular direction or data subset. POLCA works by decomposing and disaggregating the loss to discover these sharp changes. *Left:* Loss $L(x; \theta)$ changes as the parameter setting $\theta$ moves in a low-rank training subspace. The loss is sigmoidal on each axis, with differently timed inflections along basis vectors $b_1$ and $b_2$. These breakthroughs disappear in the smooth sum of the sigmoids which represents the exact loss. *Right:* The average of sigmoidal functions—including loss along basis vectors $b_1$ and $b_2$—elides individual breakthroughs. The more differently-timed breakthroughs underlie the loss, the more hidden each breakthrough is.

breakthroughs; such an example may undergo changes synchronized with different data subsets at different times. Furthermore, distinct concepts might appear simultaneously, erroneously merging their data clusters. To recover the underlying learned concepts, we may have to identify multiple separate breakthroughs which affect the loss curve of a single example.

To disentangle these effects for a single sample, we separate the optimization space into specific gradient directions. When the loss changes during training, it is the result of movement across all parameters in a high-dimensional space. We **decompose** this loss change from an **exact** trajectory in the full-rank parameter space into a collection of movements along each dimension. By analyzing these loss curves along specific basis vectors, we identify conceptual breakthroughs that rely on particular directions of movement. The latter analysis permits further granularity in clustering data, as final performance on an individual example may rely on multiple conceptual breakthroughs, each corresponding to a particular linear direction in training. In summary:

- We introduce a modified form of Loss Change Allocation (Lan et al., 2020) called **Projection Oriented Loss Change Allocation (POLCA)** to measure changes in loss due to parameter adjustments in arbitrary directions during training (Section 3.2).

- We show that some learned concepts can be identified by clustering exact loss, while others cannot (Section 4.2). Using POLCA, we extend our cluster analysis to identify these hidden conceptual breakthroughs obscured in the exact loss curves. We automatically identify specific concepts learned during breakthroughs in both synthetic (Section 4) and natural language settings (Section 5).

## 2 BACKGROUND: HOW MUCH CAN WE LEARN FROM LEARNING DYNAMICS?

Various loss breakthroughs have been interpreted as learning specific concepts. But why expect additional interpretable breakthroughs to underlie periods of undifferentiated, gradual model improvement? Our approach is justified by the nature of the loss surface's complexity, illustrated by Figure 1, in which a smooth curve emerges by eliding breakthroughs from each dimension.

**Why expect multiple breakthroughs?**   A very early phase transition is to be expected early in training after a brief memorization stage (Shwartz-Ziv & Tishby, 2017). In this sense, the most celebrated breakthroughs—those signaling the formation of induction heads or arithmetic grokking—are not the first drops in their training curves. Some breakthroughs even depend on earlier breakthroughs, as observed in synthetic tasks (Abbe et al., 2021) and in grammar acquisition (Chen et al., 2024a). If one concept depends on another, each must appear at a different timestep, requiring multiple breakthroughs. Furthermore, as shown by Saxe et al. (2019), summing many phase transitions can result in a smooth curve, supporting our hypothesis that these breakthroughs can appear in stable regions of the loss curve.

Multiple breakthroughs can also come from differences in gradient scale along different directions. Ma et al. (2022) even attributed the early edge-of-stability phase transition (Jastrzębski et al., 2020; Cohen et al., 2022) to multiscale structure of the loss surface and, furthermore, noted that this multiscale structure emerges at the range where models become *singular*: their loss lacks a quadratic approximation in terms of model parameters, creating conditions for breakthroughs under Singular Learning Theory (Watanabe, 2010; Wei et al., 2020; Wang et al., 2024). They argued that this structure is the product of both nonuniform data and nonconvex objectives, respectively justifying the *disaggregation* and *decomposition* which we apply to interpret training dynamics.

**Why disaggregate the aggregate loss?**   We track learning on training datapoints and subpopulations, rather than the whole training set, because relevant skills can be acquired at different rates (Arora & Goyal, 2023; Chen et al., 2024b). Individual samples thus exhibit changes in loss out of line with the monotonic average trend (Xia et al., 2023; Rosenfeld & Risteski, 2024). In full-batch gradient descent, Cohen et al. (2022) identified non-monotonicity arising from oscillation about the maximum Hessian eigenvector. Rosenfeld & Risteski (2024) demonstrated that these oscillations occur across different axes for different samples and identified the primary cause: surprisingly human-interpretable semantic features. Even when the loss seems stable, performance can oscillate on edge cases until the model develops relevant capabilities (Qin et al., 2024; Bhaskar et al., 2024). We hypothesize that oscillation represents competing skills that are relevant to different subsets of data. To test this hypothesis, and to interpret the meaning of these directions, we disaggregate the loss into clusters with similar dynamics.

**Why decompose the exact loss?**   Michaud et al. (2024) analyzed the scaling behavior of models with respect to individual tokens and identified a limitation of token-wise analysis of breakthroughs, which they called *polygenic* scaling effects—samples which combine multiple skills and therefore exhibit breakthroughs at multiple scales. Our POLCA decomposition directly addresses this limitation by decomposing the loss of each token along multiple basis vectors. If we assume that a specific skill is enabled by movement along that skill's basis vector, then the loss change attributed to movement along that vector will accelerate at the moment the skill is acquired, for every sample that requires that skill. In this manner, the sample transitions from early to late dynamics through a basis-specific loss breakthrough. In other words, by monitoring changes in directions corresponding to specific skills, we support the speculation of Nanda et al. (2023) that "*phase transitions are everywhere.*"

**Why is linear decomposition sufficient?**   In practice, a conceptual breakthrough might not occur in a single direction that persists throughout training. However, there is abundant evidence that linear bases of the low-rank (Gur-Ari et al., 2018) training subspace are conceptually meaningful. In the late stages of training, loss is convex on the line connecting a pair of checkpoints (Frankle et al., 2020) if those checkpoints express similar capabilities (Juneja et al., 2023) and mechanisms (Lubana et al., 2023). If a pair of high-dimensional models lack this linear connection, they still connect nonlinearly (Draxler et al., 2019); however, while parameter settings sampled from their *linear* connections improve broadly on the capabilities of the original models, those sampled from their *nonlinear* connections are less robust than the originals (Juneja et al., 2023, ref Appendix G). These observations suggest that linear decomposition should preserve meaningful conceptual features on the loss surface, and our experiments show that the resulting directions are interpretable in practice.

## 3   METHODS

The key to our approach is the separate consideration of each example's **datapoint loss** changes throughout training. We contrast this individualized metric with **aggregated loss** across an entire

dataset. Using the datapoint loss, we can cluster individual example $x$ on the basis of its loss $L(x; \theta_t)$, change in loss $L(x; \theta_t) - L(x; \theta_{t-1})$, or magnitude of change $|L(x; \theta_t) - L(x; \theta_{t-1})|$ during training.

We next decompose the loss itself into specific directions in the weight space, motivated by several considerations: First, while we have moved from an aggregated loss metric to a more granular datapoint loss metric, we are still only considering breakthroughs that are general enough to be perceived in loss curves. Second, an individual datapoint may benefit from a variety of conceptual breakthroughs, but will not be clustered on the breakthroughs individually. Finally, once we have identified a subset of the data as benefiting from a particular conceptual breakthrough, decomposing into individual weight directions allows us to locate where in the weights the breakthrough occurs and to thereby identify the mechanism involved.

When we break the **exact loss** curve down by directional movement during training, the resulting **decomposed loss** will reveal breakthroughs that are specific to a given direction. Our procedure follows three steps: (1) select a basis, (2) decompose the loss along that basis to highlight particular learning events, (3) cluster datapoints according to their shared learning events.

## 3.1 FINDING THE BASIS

---

**Algorithm 1** Finding the decomposed optimization basis

---

**input:** Training set $X$, Model checkpoints $\{\theta_t\}_{t=1}^T$.
$B_0 \leftarrow \emptyset \in \mathbb{R}^{d \times 0}$.
**for** $t = 1 \ldots T$ **do**
    $\Pi_\perp \leftarrow I - B_{t-1}(B_{t-1}^\top B_{t-1})^{-1} B_{t-1}^\top$
    $\mathcal{H} \leftarrow \nabla_\theta^2 \mathcal{L}(X, \theta)$.
    Define $B^+ \in \mathbb{R}^{d \times k}$ as the top $k$ eigenvectors of $\Pi_\perp \mathcal{H}$ (e.g., via the Lanczos method).
    $B_t \leftarrow [B_{t-1}, B^+]$.
**end for**
**return** $B_T$

---

To decompose the loss, we first require an interpretable orthogonal basis. We efficiently compute the eigenvectors of the Hessian matrix using CoLA (Potapczynski et al., 2023) to construct a restricted training subspace. We expect this basis to be interpretable because each basis vector captures a large gradient covariance and therefore represents a potential decision boundary. We select this basis for interpretability, but our approach can use an arbitrary choice of basis—for example, one which targets a particular use case or efficiently leverages optimizer preconditioner values.

The basis is constructed as shown in Algorithm 1. Given $T$ intermediate training checkpoints and a number $k$ of eigenvectors to compute at each checkpoint, we seek a low rank $Tk$-dimensional subspace which captures most of the movement during optimization (Gur-Ari et al., 2018). We construct this basis iteratively, starting with $B_0 = \emptyset$: at each checkpoint $t$, we take checkpoint weights $\theta_t \in \mathbb{R}^d$ and project their loss Hessian onto the nullspace of $B \in \mathbb{R}^{d \times (t-1)k}$. From the resulting projection, we append the top $k$ eigenvectors to $B_{t-1}$. We compute the eigenvectors using Hessian-vector products Golmant et al. (2018) to avoid explicitly constructing the full Hessian matrix. The resulting basis is designed to include directions of highest curvature at each checkpoint so that it will capture synchronized loss behavior throughout training. Note that the very top eigenvectors are likely to reflect local oscillation, rather than conceptually meaningful long-term movement (Song et al., 2024), but as we continue to add to the low rank basis, we include more directions of long-term stable movement. We discard the oscillatory directions which do not provide an overall decrease in loss over the course of training according to POLCA by removing the directions with an increase in the mean projected loss from checkpoint 1 to $T$. In this manner, we first construct a basis based on local information, then filter out directions that do not represent long-term movement. This construction finds local high-curvature directions that may be important for breakthroughs in the intermediate stages of training while ensuring that the basis does not overfit to local oscillations.

## 3.2 DECOMPOSING THE LOSS WITH POLCA

To decompose the loss along our basis, we propose a modified version of Loss Change Allocation (LCA) (Lan et al., 2020) which we call Projection-Oriented Loss Change Allocation (POLCA). LCA is a tool for analyzing changes in aggregated loss on dataset $X$ between two checkpoints. The output of LCA is the empirical loss change between a pair of checkpoints which can be attributed to the motion of each individual weight unit. Given two consecutive checkpoints with parameters $\theta_t$ and $\theta_{t+1}$, LCA reformulates the change in loss as its first-order Taylor approximation, a sum of the loss changes attributed to the movement of each individual model parameter $\theta^{(j)}$:

$$L(X; \theta_{t+1}) - L(X; \theta_t) \approx \sum_{j=0}^{d} (\nabla_\theta L(X; \theta_t))^{(j)} (\theta_{t+1}^{(j)} - \theta_t^{(j)}) = \sum_{j=0}^{d} LCA(X; \theta^{(j)}) \tag{1}$$

The POLCA decomposition differs from LCA in three key ways. First, we do not restrict each direction to correspond to a single unit $\theta^{(j)}$, instead permitting an *arbitrary* orthonormal basis vector $b \in B_T$ to replace the axis-aligned basis vectors in LCA; we project onto this basis vector using the dot product $\langle b, \cdot \rangle$. Second, we are interested in changes in the loss on each individual example $x \in X$, not the entire dataset $X$. These first two modifications provide the first-order POLCA decomposition:

$$L(X; \theta_{t+1}) - L(X; \theta_t) = \sum_{x \in X} L(x; \theta_{t+1}) - L(x; \theta_t)$$

$$\approx \sum_{x \in X} \sum_{b \in B_T} \langle b, \nabla_\theta L(x; \theta_t) \rangle \langle b, \theta_{t+1} - \theta_t \rangle \tag{2}$$

The third key difference is that we use a second-order approximation because this basis is constructed explicitly from the Hessian eigenvectors. To understand why this choice of basis warrants a second-order approximation, recall that each basis vector $b$ is an eigenvector of the Hessian matrix $\mathcal{H}_{t'}(X)$ at some timestep $t'$, where $b$ is chosen because it has the largest eigenvalue $\lambda_{t'}(X, b)$ over the whole dataset. If we assume that the top eigenvectors of the *aggregate* Hessian maintain high curvature at other points in training and on *individual* datapoints, then the scaling factor in the second-order Taylor term will be very large even at the datapoint level. Limiting the approximation to only the first order term gives poor guarantees on error, as the second-order term could be expected to dominate. Although empirically the difference between the first and second-order values is small (see Appendix I), we nonetheless guarantee a better estimate due to lower Lagrange error bounds by computing the second-order approximation below.

Exact computation of the second-order term would be intractable, requiring computation of the top eigenvalues/vectors for each individual datapoint $x$. Instead, we can approximate it by substituting the true eigenvalue, denoted $\lambda_t(X, b) := b^\top \mathcal{H}_t(X) b$, with the curvature of the individual loss in the direction $b$, i.e. $\lambda_t(x, b) = b^\top \mathcal{H}_t(x) b$. If the aggregate Hessian eigenvector $b$ is close to the span of the top eigenvectors of the datapoint-specific Hessian for $x$, this provides a reasonable estimate while reducing calculation to a single Hessian-vector product per eigenvector. We therefore approximate the basis projection of the datapoint Hessian $h(x, b, \theta_t)$ as derived in Appendix C.

$$h(x, b, \theta_t) = \frac{\lambda_t(x, b)}{2} \langle \theta_{t+1} - \theta_t, b \rangle^2 \tag{3}$$

$$\approx \frac{\lambda_t(X, b)}{2} \cdot \langle \theta_{t+1} - \theta_t, b \rangle^2 \times \frac{\langle L(x; \theta_{t+1}) - L(x; \theta_t), b \rangle}{\langle L(X; \theta_{t+1}) - L(X; \theta_t), b \rangle} \tag{4}$$

$$= \tilde{h}(x, b, \theta_t) \tag{5}$$

Equipped with this second-order approximation of the datapoint Hessian's projection onto our basis, we account for the high curvature and possible domination by the higher order term by modifying Equation 2 into the second-order Taylor expansion using the approximation from Equation 5. We can compute this second-order term with limited additional computational complexity by keeping track of the eigenvalues for each Hessian eigenvector and the aggregate gradient at each checkpoint.

$$L(X; \theta_{t+1}) - L(X; \theta_t) \approx \sum_{x \in X} \sum_{b \in B_T} \langle b, \nabla_\theta L(x; \theta_t) \rangle \langle b, \theta_{t+1} - \theta_t \rangle + \tilde{h}(x, b, \theta_t) \tag{6}$$

$$= \sum_{x \in X} \sum_{b \in B_T} POLCA(x, b; \theta_t) \tag{7}$$

### 3.3 CLUSTERING THE LOSS

POLCA, above, provides curves that show how a decomposed loss changes with respect to each training example. We assume that if several examples show similarly timed loss changes in the same direction, they likely rely on the same conceptual breakthroughs or learning events; therefore, they are likely to share a required skill, or specific capability needed for a given task. We cluster POLCA training histories to recover these skill groups. For each datapoint $x$, we compute the total cumulative change in loss along each basis vector $b$ by summing over the previous POLCA values. We denote this sum the **projected loss** $L_b(x, \theta_t)$.

$$L_b(x, \theta_t) = \sum_{i=0}^{t-1} POLCA(x, b; \theta_i) \tag{8}$$

We obtain 1d projected loss trajectories for breakthrough clustering by computing $L_b(x, \theta_t)$ at every time $t$. We cluster trajectories using Hierarchical Density-Based Spatial Clustering of Applications with Noise (HDBSCAN) Campello et al. (2013) because it distinguishes cluster outliers and discovers clusters with variable density (i.e., similarly shaped curves that lay far apart in their metrics). We cluster the trajectories for each basis vector separately to ensure that the clustering can capture multiple skills per token.

### 3.4 DEFINING AND IDENTIFYING HIDDEN BREAKTHROUGHS

We use POLCA to recover hidden breakthroughs in training, so we must quantify whether clustered trajectories are distinguished by breakthroughs. We use the formulation defined by Chen et al. (2024b) to compute the start of a breakthrough in a given function $f$ for a given datapoint $x$ and basis vector $b$:

$$\text{break}(f, x, \Delta) = \arg\max_t [f(x, t + \Delta) - f(x, t)] - [f(x, t) - f(x, t - \Delta) \tag{9}$$

Here, $\text{break}(f, x, \Delta)$ approximates the maximum point of acceleration of $x$ in $f$. $f$ can be either the projected loss $L_b$ for a given basis vector $b$ or the exact loss $L$. We define a *hidden breakthrough* as a breakthrough that occurs in the flat region of the exact loss curve. That is, if we set a threshold $\tau$ beyond which the exact loss curve is flat, then a given set $X' \subseteq X$ has a hidden breakthrough in a metric $f$ if the expected value of the start of breakthroughs in $X'$ is greater than $\tau$:

$$\text{hidden}(f, X', \Delta) = \mathbf{1} \left\{ \mathbb{E}_{x \in X'} \left[ \arg\max_t [f(x, t + \Delta) - f(x, t)] - [f(x, t) - f(x, t - \Delta)] \right] > \tau \right\} \tag{10}$$

## 4 ARITHMETIC LANGUAGE MODELING

We validate our POLCA clustering method in a synthetic setting using an arithmetic addition task. Our clusters reflect categorical concepts within the data, even when those concepts are not discoverable by clustering directly on loss curves. Specifically, if we cluster on exact loss curves, we recover digit positions—but if we instead cluster on POLCA curves, we *also* recover the skill of "carrying" a digit.

### 4.1 EXPERIMENTS

**Data** Our synthetic experiments use data from the arithmetic addition setting in Chen et al. (2024b), where the model is trained to compute the sum of two 3-digit numbers. This setting has 4 skills corresponding to each of the digits in the output sum. Note that the digit in the 1000s place is always a `<0>` or `<1>` token since the two input summants are 3 digits long. As shown in Appendix Figure 6 and Chen et al. (2024b), the skills corresponding to the digits have different loss curves, so we will easily recover the digit skill categories by clustering exact loss curves.

Unlike our source material, we also consider an additional skill: arithmetic carries to the output token (Figure 2). This skill corresponds to the case where instead of simply adding the two tokens at the corresponding digit in the input, the model must also carry a 1 from the previous digit. Digit-specific addition skills lead to clearly distinguishable exact loss curves, whereas carrying skills do not (Appendix Figure 7)—but carries will become clear on our decomposed gradient basis. We provide additional skill and labeling details in Appendix E.1.

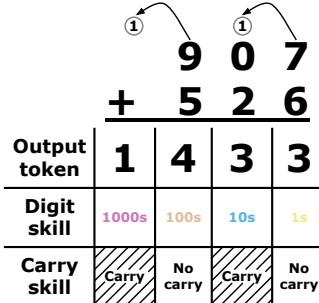

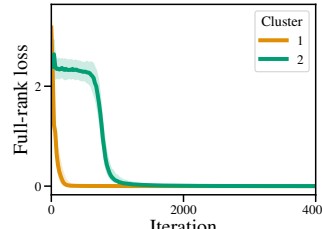

(a) Mean exact loss trajectories per cluster.

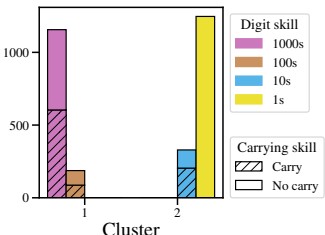

(b) Arithmetic skill composition of the clusters.

Figure 2: **Diagram of arithmetic addition task.** An example of 3-digit addition, labeled with the skills required for each of the output tokens.

Figure 3: **Exact loss trajectory clustering on the arithmetic task.** We use HDBSCAN to cluster the exact loss trajectories. This approach, unlike our POLCA clustering method, fails to recover clusters associated with the carrying skill (the maximum fraction of carries is 0.51).

**Setup details** We train a 3-layer (9 million parameter) Transformer model with embedding dimension 512, 4 attention heads per layer, and an MLP dimension of 2048, following prior work (Olsson et al., 2022a). We study a validation set with 1250 data points and 5000 output tokens throughout training. We compute the loss and POLCA values for each token at each interval of 20 training steps. The POLCA basis uses the eigenvectors of the Hessian, estimated using a 1250 data point sample of the training set as detailed in Algorithm 1. We compute one new basis vector every 200 training steps, for a total of 50 basis vectors. We provide further ablations on the decomposition strategy and choice of POLCA basis in Appendices G and H. We train the model for 10000 steps, but trim the x-axis of the plots at 4000 to better display the breakthroughs.

**Clustering** As described in Section 3.1, some of the top Hessian directions may represent directions of oscillation (and not learning) during training. To ensure that we are investigating directions where the model is learning on average, we only consider the basis vectors for which the mean projected loss decreases. Then for each remaining basis vector, we remove all of the tokens for which the decomposed loss increases, therefore retaining only tokens which rely on the vector during training. This removes 2360.8 out of 5000 tokens on average—suggesting that any given direction is irrelevant to learning many examples. We use HDBSCAN to cluster the remaining tokens, discarding the tokens it marks as outliers. Through this process, we find subpopulations of the data that have similar projected loss trajectories.

## 4.2 RESULTS

**Comparison to the exact loss** In our clustering experiments on arithmetic addition skills, we first consider whether decomposition is necessary for identifying conceptual skills. As a baseline, we therefore cluster tokens solely on their exact loss curves, rather than their decomposed loss curves. According to the exact loss clustering results in Figure 3, we can recover—to a substantial degree—the *digit* skill by clustering only on the exact loss, likely because the digits have very different loss trajectories. However, as shown in Figure 3 and Table 1, we cannot recover clusters that are homogenous with respect to the *carrying* skill from the exact loss alone.

| Decomposition strategy | Maximum carry homogeneity | Clusters with hidden breakthroughs |
|---|---|---|
| Exact loss | 0.514 | 0.0 |
| Change in exact loss | 0.524 | 0.0 |
| LCA (Lan et al., 2020) | 0.792 | 0.019 |
| POLCA | 0.973 | 0.355 |

Table 1: **Cluster quality comparison.** We compute the maximum fraction of points within all clusters that contain a carry for the specified digit and the fraction of clusters with hidden breakthroughs past the plateau in the exact loss at $\tau = 1000$. For details and other metrics, see Appendix G.

**Recovering concepts with POLCA clustering** Because exact loss clustering failed to recover the carrying skill, we will recover it with a different clustering method. Instead of exact loss, we will cluster on each basis vector's projected loss using the POLCA decomposition. The projected loss value $L_b(x, \theta_t)$ (Equation 8) represents the cumulative loss change of $x$ attributed to movement

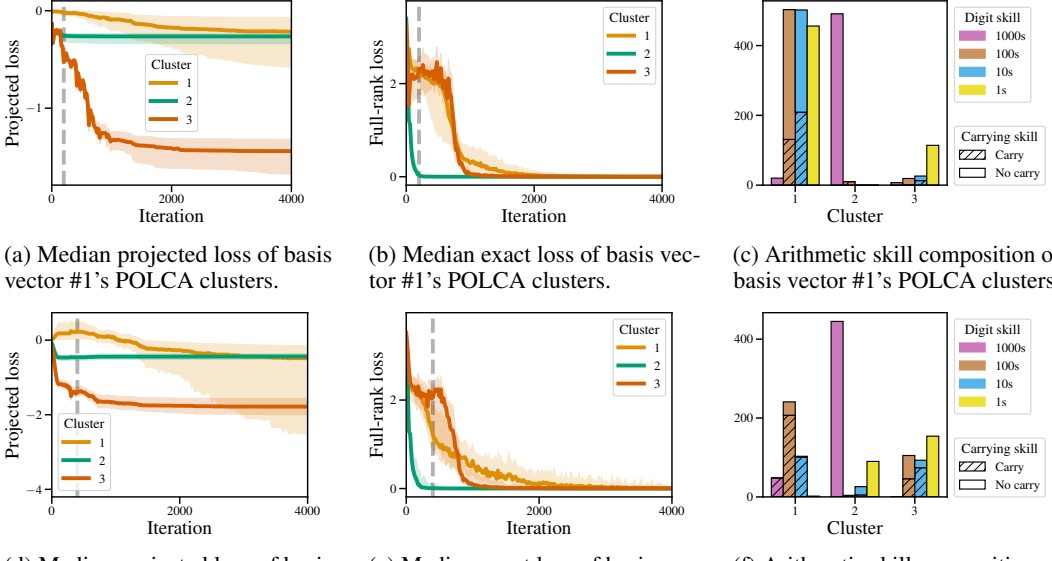

(a) Median projected loss of basis vector #1's POLCA clusters.

(b) Median exact loss of basis vector #1's POLCA clusters.

(c) Arithmetic skill composition of basis vector #1's POLCA clusters.

(d) Median projected loss of basis vector #2's POLCA clusters.

(e) Median exact loss of basis vector #2's POLCA clusters.

(f) Arithmetic skill composition of basis vector #2's POLCA clusters.

Figure 4: **Arithmetic data clusters with POLCA.** We perform POLCA clustering on the top 2 basis vectors, and report the cluster medoid and quartiles (*left*), median exact loss (*center*), and cluster skill composition (*right*) for each basis vector in order. Vertical lines mark the timestep when the relevant basis vector was sampled; note that a vector's breakthroughs are not directly associated with this timestep. We find that the first basis vector recovers the digit skill whereas the second basis vector recovers the carrying skill (cluster #1 has homogeneity 0.90). The clusters computed from the POLCA trajectories show changes in the decomposed loss that are obscured in the exact loss curves.

along basis vector $b$. By clustering the projected loss trajectories, we find that the top 2 basis vectors produce two easily-described and near-homogeneous clusters: one which contains examples of the 1000s place digit and one which contains examples of arithmetic carrying for all digits (Figure 4 Appendix Figure 8). We therefore determine that POLCA clustering recovers subtler skills—like carrying—that are challenging to reconstruct from the exact loss or parameter-aligned LCA curves alone (Table 1). In Appendix Table 9, we also use Equation 10 to compare the fraction of clusters with hidden breakthroughs past step 1000 (with $\Delta = 100$), where the mean exact loss plateaus (Appendix Figure 6), and find in Table 1 that POLCA discovers the highest fraction of clusters with hidden breakthroughs.

Figure 4 shows POLCA clustering for the first two basis vectors. Along these basis vectors, certain data subpopulations show changes in the projected loss (Figures 4a and 4d) that do not occur as visibly in their smoother exact loss curves (Figures 4b and 4e). *We conclude that arithmetic carries rely on breakthroughs along specific dimensions during training, but these breakthroughs may be elided in the exact loss curve.*

## 5 ENGLISH LANGUAGE MODELING

We apply our approach to a real-world causal language modeling task and show that POLCA breakthrough clustering recovers interpretable conceptual skills in the natural language setting.

### 5.1 EXPERIMENTS

For the natural language modeling setting, we use the English Wikipedia dataset (Wikimedia Foundation, 2022) from March 2022 to train a 3-layer (40m parameter) model. We use the same POLCA setup as in the arithmetic addition setting (see Appendix E.2 for details). As in the arithmetic setting, we only consider directions where the projected aggregate loss overall decreases to filter out directions of oscillation. We also discard the datapoint-specific POLCA trajectories along the directions where

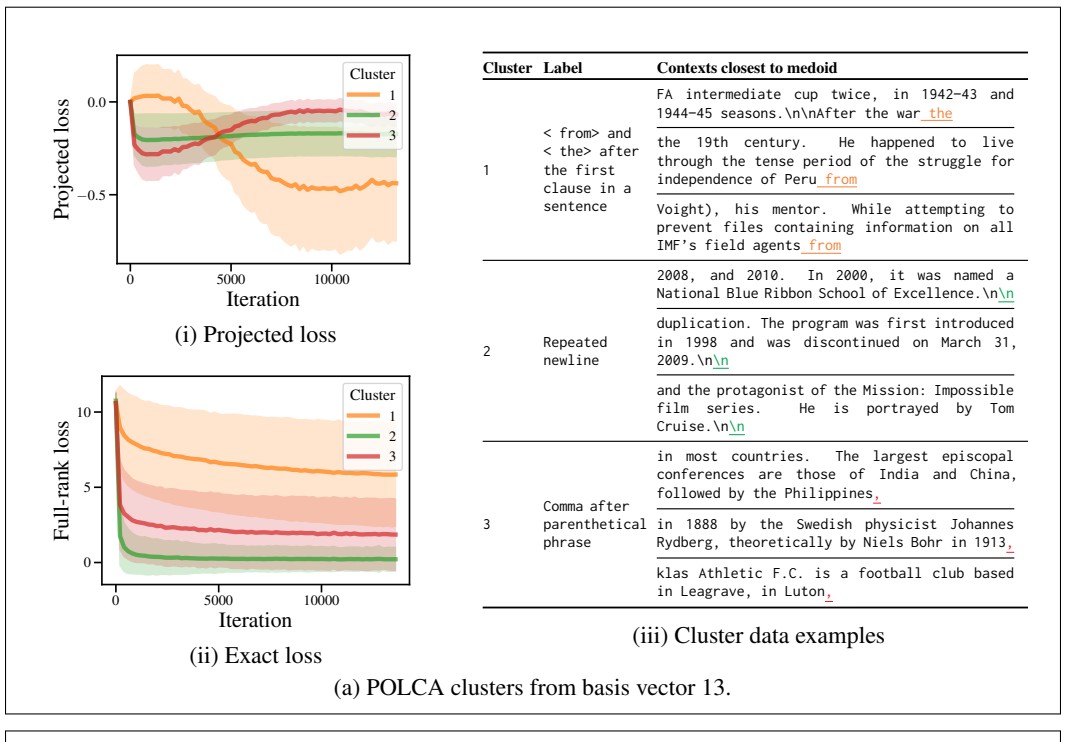

(a) POLCA clusters from basis vector 13.

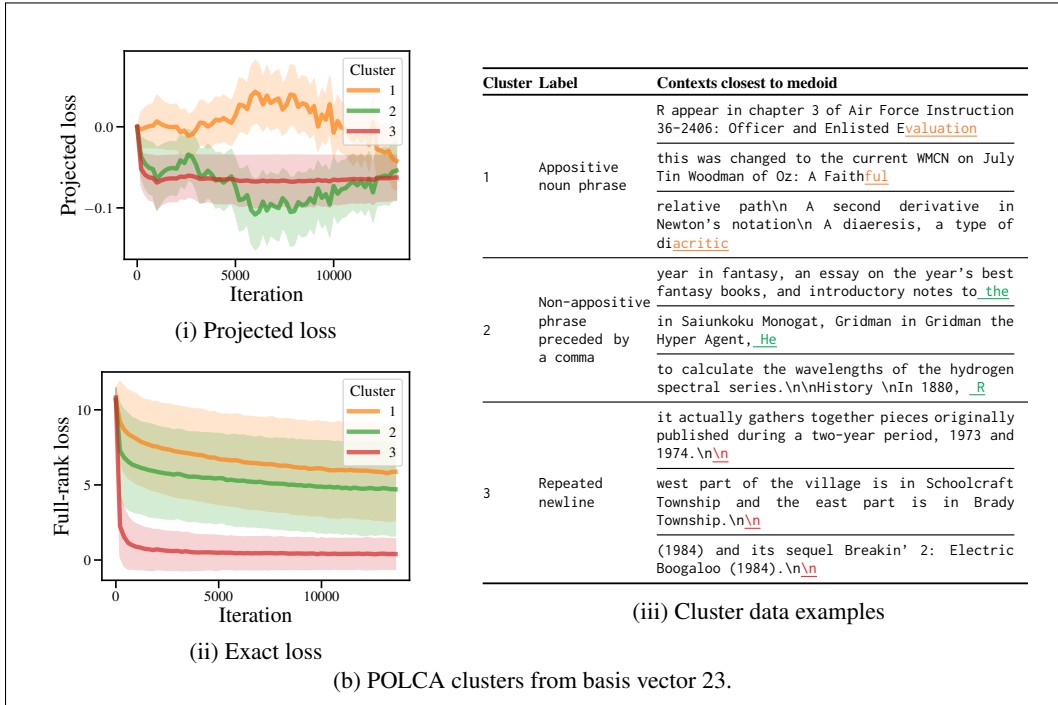

(b) POLCA clusters from basis vector 23.

Figure 5: **Examples of English LM data clusters with POLCA.** After clustering on POLCA trajectories for two illustrative basis vectors, we report their average decomposed POLCA trajectories (5a(i) and 5b(i)). Figures 5a(ii) and 5b(ii) show the average of the exact loss trajectories for each of the POLCA trajectory clusters. For each cluster, we provide a label based on the top POS tags and tokens in the cluster and the top 10 contexts closest to its medoid. We report the 3 contexts closest to the cluster medoid and color the corresponding token. Clustering on the decomposed POLCA trajectories reveals low-rank breakthroughs at times when the full-rank exact loss curve remains smooth. See other examples in Appendix K.

the token's decomposed loss increases, which removes an average 6655.5 out of 12600 tokens per direction. After clustering the remaining token trajectories, we discard HDBSCAN-marked outliers.

**Automatic labeling.**    To analyze the concepts represented by each cluster, we look for syntactic and lexical patterns shared by the cluster data.[1] Our automatic labels are based on the target token and its preceding trigram. To obtain automatic labels for a cluster, we compute the frequency of each POS tag (tagged by spacy) in it. We then automatically label each cluster with the smallest set of POS tags required to compose 70% of the cluster's 4-gram samples. For example, if over 70% of the token instances in a cluster are preceded by a comma `<,>`, the cluster would be automatically labeled as `1 token after PUNCT`. We consider basis vectors with at least one cluster with a *simple* label—one including at most two POS tags (not counting `<PAD>` tokens). Starting with 30 basis vectors, we remove 4 because the average decomposed loss increases. We find 22 of the remaining basis vectors have at least one cluster with a simple label as defined. We refine these automatically assigned labels by examining the most frequent tokens in the cluster and the ten examples closest to the cluster medoid and manually selecting a label that is consistent with the automatic POS label and the ten closest examples to the medoid. See Appendix E.2 for further labeling details.

## 5.2    RESULTS

On language models, POLCA clustering again reveals hidden breakthroughs along each basis vector. We show a selection of clusters from specific basis vectors in Figure 5 and others in Appendix K (Figures 10 and 11). Clustering projected loss trajectories along each basis vector, we find groups that correspond to various grammatical constructions. Some show apparent breakthroughs in their projected loss, like the cluster corresponding to predicting `< to>` and `< from>` after the first clause in a sentence (Figure 5a(i)). We also observe clusters whose projected loss trajectories move in opposing directions along certain basis vectors. For instance, in Figure 5b, cluster 1 contains appositive noun phrases and cluster 2 has syntactically similar—but non-appositive—noun phrases (such as list items). These clusters visibly mirror each other's movement—though the clustered decreases in loss are generally larger than their opposing cluster's increases in loss.

Figure 5 shows that despite their smooth *exact* loss curves, POLCA clusters have sudden changes in their *decomposed* loss curves at different points during training. Clusters from the exact loss curves, by contrast, do not reveal breakthroughs except very early in training (Appendix Figure 9). We conclude that POLCA reveals breakthroughs in the decomposed loss that are obscured in the exact loss. Through clustering, POLCA can explain how different skills are learned during training.

## 6    CONCLUSIONS

This work introduces POLCA clustering, a method to identify learned skills from decomposed loss trajectories. POLCA decomposes the loss on two levels: individual datapoints and specific directions in the weight space. We use this decomposition to discover clusters that share breakthroughs obscured by loss metrics. In language modeling and synthetic settings, these clusters recover interpretable skills which appear to emerge at particular moments during training.

These are promising findings for meaningfully interpreting large models. By recovering breakthroughs in identifiable skills, we support the hypothesis that high-dimensional learning typically entails a series of breakthroughs at various scales. When a breakthrough appears in training, it suggests a naturally discrete category; the model either knows the concept or doesn't know it, with little middle ground. Humans think in categorical concepts, so these breakthroughs are far more interpretable than the continuous data interpolations often assumed in learning theory.

---

[1]In principle, our method can uncover skills that are not describable through these templates, but templating allows automatic labeling. Our templating approach is similar to the automated explainer tool N2G (Foote et al., 2023), a popular ngram-based evaluation metric for unsupervised interpretability methods (Gao et al., 2024).

## REPRODUCIBILITY STATEMENT

We implement our models and experiments using open-source libraries and datasets. We provide detailed hyperparameters in Appendix D and a thorough explanation of the experimental setup in Section E. Our code is available at `https://github.com/skangasl/POLCA`.

## ETHICS STATEMENT

This work provides a method for better understanding the training dynamics of language models. The trained models may contain biases from the training datasets.

## ACKNOWLEDGEMENTS

This work was informed by helpful conversations with Nikhil Vyas, Nicholas Lourie, Mike Lepori, and Ekdeep Singh Lubana. This material is based upon work supported by the National Science Foundation Graduate Research Fellowship under Grant No. DGE 2140743. Any opinion, findings, and conclusions or recommendations expressed in this material are those of the authors(s) and do not necessarily reflect the views of the National Science Foundation. This work was enabled in part by a gift from the Chan Zuckerberg Initiative Foundation to establish the Kempner Institute for the Study of Natural and Artificial Intelligence.

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

## A    LIMITATIONS AND FUTURE WORK

Our method of constructing a basis is inspired by the existing literature on training in restricted subspaces, but represents an obvious site of improvement. The top eigenvectors of the Hessian, like the axis-aligned basis, could represent many concepts in superposition. Therefore, some non-orthogonal basis might represent interpretable concepts more cleanly than our orthogonal basis, though it would no longer provide a low-rank decomposition. Furthermore, our basis is constructed by local curvature and then filtered to favor directions of long-term movement; other bases may favor long-term movement by construction. In general, we consider the ideal basis to be an open question.

Our experiments are limited to small models. The two main challenges with scaling POLCA are the Hessian basis computation and the frequency of checkpoints used to sample the POLCA trajectories. Scaling this work to larger models may require using a basis that is less computationally expensive to compute than Hessian eigenvectors, but our results from Appendix H indicate that this is likely possible with limited impact on the cluster quality. The small scale of models that we use in our experiments allows for very high granularity of checkpoints used to compute both the basis and the POLCA trajectories. For larger models, this may be computationally infeasible and a lower checkpoint frequency may be needed, resulting in less signal for clustering the POLCA trajectories.

Our current experiments are limited to language models. However, in principle our approach is model-agnostic and can be applied to any deep neural network. Applying POLCA to other modalities is an exciting direction for future work.

The labeling approach that we use in the natural language setting relies on POS tagging. This labeling strategy allows for unsupervised, automatic identification of these syntactic skills and ensures strict interpretable labels. However, it fails to capture many human-interpretable language modeling skills. The discarded vectors may (and likely do) contain other interpretable skill clusters that are not found by automated labeling.

## B    STATEMENT ON USE OF LARGE LANGUAGE MODELS

We used generative AI for debugging and minor grammar edits when writing. The authors made all significant contributions to the research, analyses, and writing.

## C    DERIVATION OF APPROXIMATE SECOND-ORDER TERM

We can approximate the difference between the gradient at time $t$ and $t+1$ as

$$
\begin{align}
g_{t+1}(X) - g_t(X) &\approx \mathcal{H}_t(X)(\theta_{t+1} - \theta_t) \tag{11} \\
\langle g_{t+1}(X) - g_t(X), b \rangle &\approx b^\top \mathcal{H}_t(X) b \langle b, \theta_{t+1} - \theta_t \rangle \tag{12} \\
&= \lambda_t(X) \langle b, \theta_{t+1} - \theta_t \rangle \tag{13}
\end{align}
$$

If we assume $b$ to also be an eigenvector of the datapoint Hessians $\mathcal{H}'_t(x)$, we can apply a similar argument for the gradient on the datapoint level.

$$
\langle g'_{t+1}(x) - g'_t(x), b \rangle \approx b^\top \mathcal{H}'_t(x) b \langle b, \theta_{t+1} - \theta_t \rangle \tag{14}
$$

Note that the assumption above (of matching Hessian eigenvectors between data points and their aggregate) is unlikely to be correct. If this assumption is violated, then the scaling factor in the following second-order Taylor term will be minuscule on the datapoint level. In practice, we have found that the second-order term has limited impact at the datapoint level (see Appendix I), but we nonetheless use it to improve our approximation. Then we may approximate it as:

$$\frac{\langle g'_{t+1}(x) - g'_t(x), b\rangle}{\langle g_{t+1}(X) - g_t(X), b\rangle} \approx \frac{b^\top \mathcal{H}'_t(x) b \langle b, \theta_{t+1} - \theta_t\rangle}{\lambda_t(X, b)\langle b, \theta_{t+1} - \theta_t\rangle} \tag{15}$$

$$\frac{\langle g'_{t+1}(x) - g'_t(x), b\rangle}{\langle g_{t+1}(X) - g_t(X), b\rangle} \approx \frac{\langle h'_t(x), b\rangle \langle b, \theta_{t+1} - \theta_t\rangle}{\lambda_t(X, b)\langle b, \theta_{t+1} - \theta_t\rangle} \tag{16}$$

$$\lambda_t(X, b) \frac{\langle g'_{t+1}(x) - g'_t(x), b\rangle}{\langle g_{t+1}(X) - g_t(X), b\rangle} \approx \langle h'_t(x), b\rangle \tag{17}$$

**Empirical Validation** We run a small-scale empirical study to test the accuracy of this estimation. For a sample of 400 tokens and the first 10 POLCA checkpoints and the top 5 basis vectors in the arithmetic setting, we compute the root mean squared error (RMSE) between the approximated second-order term $\lambda_t(X, b) \frac{\langle g'_{t+1}(x) - g'_t(x), b\rangle}{\langle g_{t+1}(X) - g_t(X), b\rangle}$ and the true second-order term $\langle h'_t(x), b\rangle$ averaged across tokens, checkpoints, and basis vectors. We find that our approximation has an RMSE of $0.145$ when compared to the ground truth second-order term, indicating that this approximation is close to the real value.

## D    HYPERPARAMETERS

In the tables below, we provide the hyperparameters used during training of the models in the synthetic arithmetic and language modeling settings. We use NVIDIA H100 80GB HBM3 GPUs for our experiments and run each training run on a single GPU.

We selected the clustering hyperparameters to maximize empirical performance in the synthetic setting and used similar values relative to the number of tokens in the natural language experiments. We chose the POLCA hyperparameters to maximize the frequency of POLCA checkpoints and basis checkpoints within computational constraints.

Table 2: Hyperparameters for training the synthetic arithmetic model

| Hyperparameter | Value |
| --- | --- |
| Number of Parameters | 9475594 |
| Steps | 10000 |
| Epochs | 1 |
| Batch size | 64 |
| Number of training tokens | 2560000 |
| Optimizer | AdamW |
| Learning rate | 1e-5 |
| Weight decay | 0.1 |
| Betas | $(0.9, 0.95)$ |
| LR Schedule | $\min(i/100, 1.0)$ |

Table 3: POLCA hyperparameters for the synthetic setting

| Hyperparameter | Value |
| --- | --- |
| Basis checkpoint interval (steps) | 200 |
| T | 50 |
| k | 1 |
| POLCA checkpoint interval (steps) | 5 |

Table 4: Arithmetic clustering hyperparameters

| Hyperparameter | Value |
| --- | --- |
| Clustering algorithm | HDBSCAN |
| Minimum cluster size | 150 |
| Minimum samples | Number of tokens / 15 |

Table 5: Hyperparameters for training the natural language model

| Hyperparameter | Value |
| --- | --- |
| Number of Parameters | 40274737 |
| Steps | 14000 |
| Epochs | 1 |
| Batch size | 64 |
| Number of training tokens | 114688000 |
| Optimizer | AdamW |
| Learning rate | 1e-5 |
| Weight decay | 0.1 |
| Betas | $(0.9, 0.95)$ |
| LR Schedule | $\min(i/100, 1.0)$ |

Table 6: POLCA hyperparameters for the natural language setting

| Hyperparameter | Value |
| --- | --- |
| Basis checkpoint interval (steps) | 750 |
| T | 30 |
| k | 1 |
| POLCA checkpoint interval (steps) | 200 |

Table 7: Natural language clustering hyperparameters

| Hyperparameter | Value |
| --- | --- |
| Clustering algorithm | HDBSCAN |
| Minimum cluster size | 300 |
| Minimum samples | Number of tokens / 15 |

# E EXPERIMENTAL DETAILS

## E.1 ARITHMETIC SETTING

**Setup.** In the arithmetic experiments in Section 4, we train a 3-layer (9m parameter) Transformer model with embedding dimension 512, 4 attention heads per layer, and an MLP dimension of 2048 (Nanda & Bloom, 2022). For a validation set with 1250 data points and 5000 output tokens, we compute the loss and POLCA values for each token at intervals of 20 steps throughout training. We compute the POLCA basis using the eigenvectors of the Hessian estimated using a 1250 data point sample of the training set. We compute a new basis vector every 200 steps for a total of 50 basis vectors.

**Labeling details.** We automatically label each token with the ground truth value of the digit and carry skills using the definition of these two skills. We label the digit skill based on the position of the

token in the output: the first token is 1000s, the second token is 100s, the third is 10s, and the fourth is 1s. We label the carry skill by computing the sum up to the next lowest digit place and determining whether it resulted in a carry to the current token. For instance, for an output in the 10s place, if the two inputs in the 1s place sum to 10 or higher, then it will be labeled with "carry", otherwise it will be labeled with "no carry".

### E.2 NATURAL LANGUAGE SETTING

**Setup.** For the natural language experiments in Section 5, we train on the English Wikipedia dataset (Wikimedia Foundation, 2022) from March 2022. We use the same POLCA setup as in the arithmetic addition setting. We train a 3-layer (40m parameter) transformer model with embedding dimension 512, 4 attention heads, and an MLP dimension of 2048 (Nanda & Bloom, 2022). We compute the loss and POLCA values for each token on a validation set of 12600 output tokens. We analyze intermediate checkpoints at intervals of 200 steps throughout training. We apply POLCA to the basis derived from the eigenvectors of the Hessian estimated using a 1000 data point sample of the training set as detailed in Algorithm 1 with $k = 1$. We compute a new basis vector every 750 steps.

**Labeling details.** We label each token and the three tokens before it using spacy for part-of-speech (POS) tagging. This produces a sequence of four POS tags as the automatic POS label for each token. To label a given cluster, we compute the frequency (across the tokens in the cluster) of each POS tag at each index. For each index in the sequence, we then automatically label the cluster with the smallest set of POS tags at that index required to make up 70% of the cluster. We then filter out any labels that require more than 2 POS tags to describe the cluster. To generate the labels reported in Section 5, we manually refine the automatically generated label by looking at the top 10 contexts closest to the medoid of the cluster. Although these manually refined labels are challenging to verify automatically, we ensure that the contexts closest to the medoid follow the manual label and that the manual label follows the automatic label generated using the POS tags.

## F EXACT LOSS TRAJECTORIES FOR THE DIGIT AND CARRY SKILLS

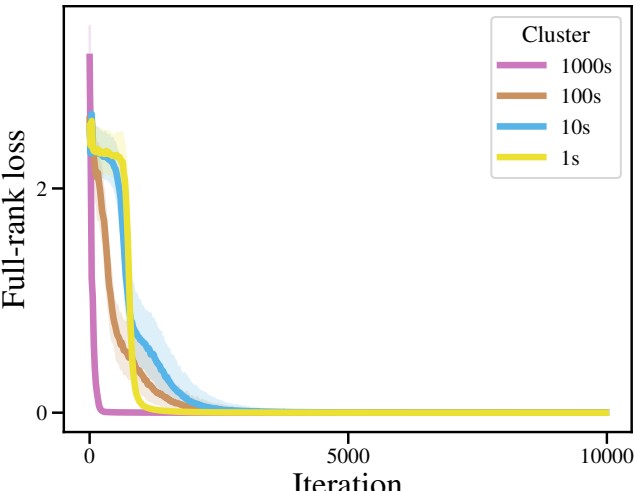

Figure 6: Median and quartiles of the loss trajectories for each digit.

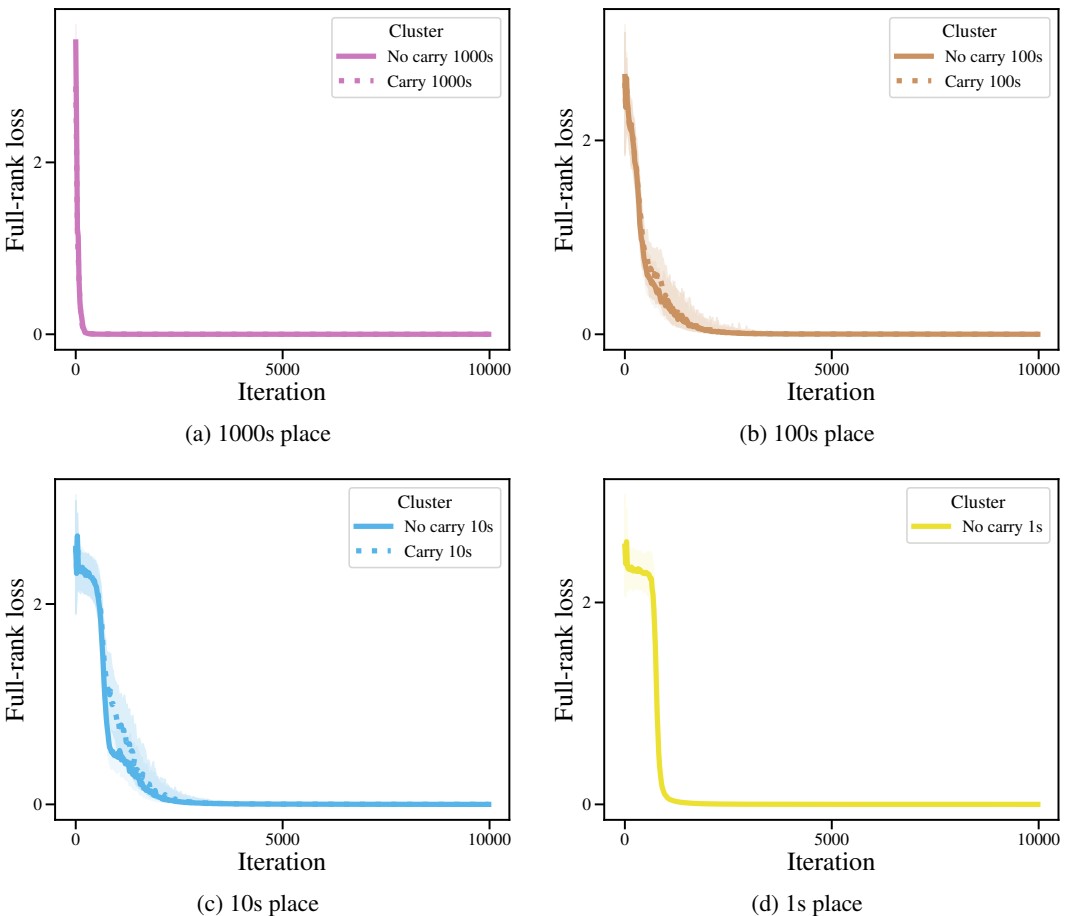

Figure 7: Median and quartiles of the loss trajectories for each digit and carry combination.

## G  DECOMPOSITION STRATEGY COMPARISON

We investigate whether POLCA is required for decomposing the loss. We expand on the results from Table 1 by showing the top three homogeneity scores for the carrying skill and adding empirical Fisher information and first order POLCA results. To compute the homogeneity score for a given basis vector, we take the maximum fraction of carries in any given cluster for that basis vector (or across all of the clusters for exact loss or change in exact loss). We then report the top three homogeneities across the full basis, as well as the recall and F1 score for the corresponding clusters. We note that we exclude HDBSCAN outliers from the recall and F1 computations. Importantly, we only consider clusters for which over $85\%$ of the tokens with the carry skill correspond to the 10s or 100s place, since carrying to the 1000s place corresponds to simply predicting a 0 or 1 in the first position (see Figure 2 for reference) and is recovered with high homogeneity for all trajectory types except exact loss.

We compare carry skill homogeneity across the following trajectory types:

- **Loss**: Exact loss trajectories.
- **Change in exact loss**: We compute the change in exact loss by subtracting the loss at checkpoint $t - 1$ from the loss at checkpoint $t$ for each timestep $t > 0$ in the exact loss trajectory.
- **Fisher information**: We approximate the empirical Fisher Information as $\|\nabla_\theta L(x, \theta_t)\|_2^2$ as in Achille et al. (2017). For each basis vector $b$, the Fisher Information projected onto $b$ is $\langle b, \nabla_\theta L(x, \theta_t)\rangle^2$.

- **Loss Change Allocation (LCA)**: We compute the datapoint-wise LCA trajectories (Equation 1) projected onto the parameters that have the top 50 highest magnitudes at the end of training.

- **First order POLCA**: We calculate the POLCA trajectory without the second order term (Equation 2)

- **POLCA**: We compute the projected loss (Equation 7).

The results from Table 8 demonstrate that POLCA finds the most homogeneous clusters with respect to the carrying skill. Moreover, POLCA and first order POLCA have comparable F1 scores. Note that many low homogeneity clusters can have high recall because they are large and thus contain most of the carry tokens while having low homogeneity.

Table 8: **Carry skill homogeneity comparison.** For each type of trajectory, we compute the fraction of points within each cluster that contain a carry to the output token and report the homogeneity, recall, and F1 score for the three clusters with highest homogeneity across all 50 vectors. POLCA recovers carry clusters with the highest homogeneity.

| Decomposition strategy | Cluster | | | |
| --- | --- | --- | --- | --- |
| | Number | Homogeneity | Recall | F1 |
| Loss | 1 | 0.514 | 0.771 | 0.617 |
| Change in exact loss | 1 | 0.524 | 0.958 | 0.678 |
| Fisher information | 1 | 0.664 | 0.947 | 0.781 |
| | 2 | 0.643 | 0.740 | 0.688 |
| | 3 | 0.637 | 0.874 | 0.737 |
| Loss change allocation (LCA) (Lan et al., 2020) | 1 | 0.792 | 0.772 | 0.782 |
| | 2 | 0.614 | 0.873 | 0.721 |
| | 3 | 0.592 | 0.626 | 0.609 |
| First order POLCA | 1 | 0.948 | 0.767 | 0.848 |
| | 2 | 0.928 | 0.769 | 0.841 |
| | 3 | 0.887 | 0.751 | 0.813 |
| POLCA | 1 | 0.973 | 0.736 | 0.838 |
| | 2 | 0.946 | 0.773 | 0.850 |
| | 3 | 0.903 | 0.762 | 0.827 |

We also compare the fraction of clusters with hidden breakthroughs for each type of trajectory. To compute whether or not a given cluster has a hidden breakthrough, we use Equation 10 with $\Delta = 100$ to identify breakthroughs past step $\tau = 1000$ where the exact loss plateaus. We find that POLCA produces the highest fraction of clusters with hidden breakthroughs. We hypothesize that this is mostly due to the basis construction, since the Fisher information and first order POLCA (both of which are computed using the same basis as POLCA) produce the next highest fraction of clusters with hidden breakthroughs.

Table 9: **Hidden breakthroughs comparison.** For each type of trajectory, we use Equation 10 to compute the fraction of clusters with a hidden breakthrough past the plateau in the exact loss at step $\tau = 1000$.

| Decomposition strategy | Fraction of clusters with hidden breakthroughs |
| --- | --- |
| Loss | 0.0 |
| Change in exact loss | 0.0 |
| Fisher information | 0.284 |
| Loss change allocation (LCA) (Lan et al., 2020) | 0.019 |
| First order POLCA | 0.307 |
| POLCA | 0.355 |

## H  POLCA BASIS COMPARISON

We test ablated bases to analyze the effect of basis choice on the POLCA breakthrough clustering. To do so, we compute the maximum carry skill homogeneities over all of the clusters when performing POLCA breakthrough clustering. We use the following bases:

- **Random orthonormal**: randomly sampled orthonormal vectors
- **Random shuffled Hessian**: basis computed using Algorithm 1, but randomly shuffling the model checkpoints
- **Top Hessian eigenvectors**: basis computed using Algorithm 1

We find in Table 10 that these ablations result in only slightly lower quality clusters with respect to homogeneity (although random orthonormal vectors have lower recall than the other two bases on average), indicating that different bases can be used for larger experiments to trade off between compute and interpretability. We also compute the fraction of clusters with hidden breakthroughs for each basis in Table 11 and find that the random orthonormal basis has a significantly lower fraction of hidden breakthroughs recovered than the other two approaches, indicating that this random orthonormal basis is not sufficient to find hidden breakthroughs late in training.

In addition to these bases, we have tested a variety of additional basis constructions, such as a stacked Jacobian, Hessian computed using a sliding window, and Hessian computed at the end of training, and chose to use the top Hessian eigenvectors since they had the best performance in the arithmetic setting.

Table 10: **Carry skill homogeneity comparison.** For each basis, we compute the fraction of points within each cluster that contains a carry to the output token and report the homogeneity, recall, and F1 for the clusters with maximum homogeneity. Using the top Hessian eigenvectors recovers slightly more homogeneous carry clusters than the other basis selection strategies. The random orthonormal vectors have high homogeneity but lower recall than the other two bases.

| POLCA basis | Cluster | | | |
| --- | --- | --- | --- | --- |
| | Number | Homogeneity | Recall | F1 |
| Random orthonormal | 1 | 0.902 | 0.655 | 0.759 |
| | 2 | 0.858 | 0.533 | 0.658 |
| | 3 | 0.838 | 0.586 | 0.689 |
| Random shuffled Hessian | 1 | 0.856 | 0.699 | 0.769 |
| | 2 | 0.852 | 0.734 | 0.789 |
| | 3 | 0.834 | 0.730 | 0.779 |
| POLCA | 1 | 0.973 | 0.736 | 0.838 |
| | 2 | 0.946 | 0.773 | 0.850 |
| | 3 | 0.903 | 0.762 | 0.827 |

Table 11: **Hidden breakthroughs basis comparison.** For each type of basis, we use Equation 10 to compute the fraction of clusters with a hidden breakthrough past the plateau in the exact loss at step $\tau = 1000$.

| POLCA basis | Fraction of clusters with hidden breakthroughs |
| --- | --- |
| Random orthonormal | 0.031 |
| Random shuffled Hessian | 0.304 |
| POLCA | 0.355 |

## I    SECOND VERSUS FIRST ORDER POLCA APPROXIMATION

Table 12: Empirical comparison of second and first order POLCA values. For the arithmetic setting, we compute the average cosine similarity and L2 distance between the second (Eq 7) and first (Eq 2) order POLCA trajectory vectors. The first and second-order approximations of the POLCA trajectories are very similar on average.

| Cosine similarity | L2 norm |
|---|---|
| 5.4891 E-4 | 0.99987 |

## J    ADDITIONAL ARITHMETIC LANGUAGE MODELING CLUSTERS

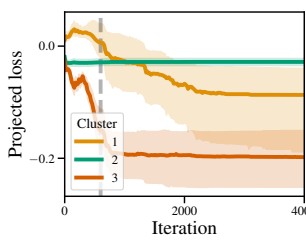
(a) Median projected loss of basis vector #3's POLCA clusters.

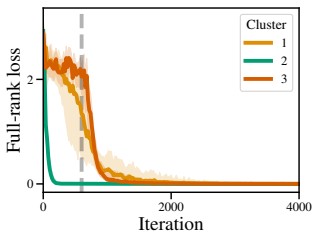
(b) Median exact loss of basis vector #1's POLCA clusters.

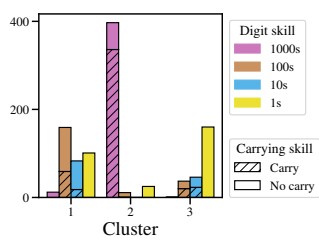
(c) Arithmetic skill composition of basis vector #3's POLCA clusters.

Figure 8: **Arithmetic data clusters with POLCA.** We perform POLCA clustering on the third basis vector, and report the cluster medoid and quartiles (*left*), median exact loss (*center*), and cluster skill composition (*right*). Vertical lines mark the timestep when the relevant basis vector was sampled; note that a vector's breakthroughs are not directly associated with this timestep. We find that the third basis vector recovers the carrying skill in the 1000s place.

## K    ADDITIONAL NATURAL LANGUAGE CLUSTERS

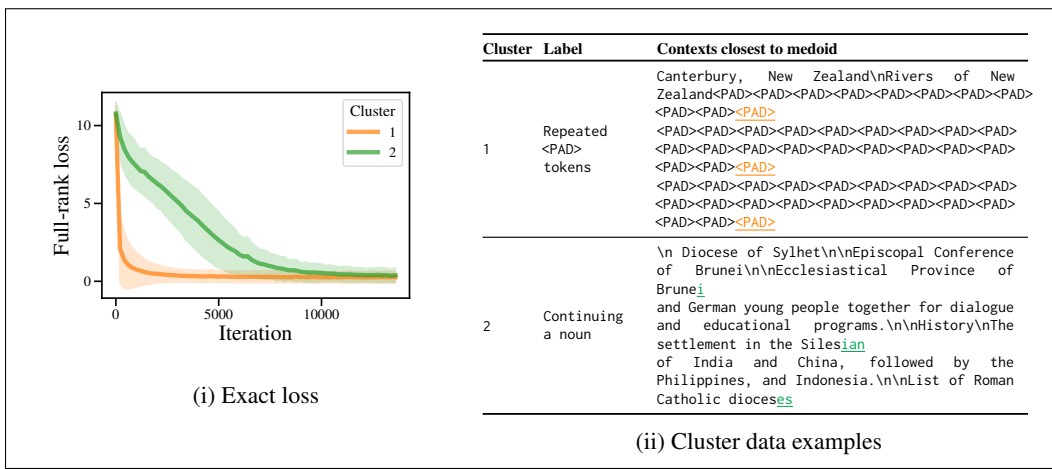

Figure 9: **English language modeling data clusters with the exact loss.** We cluster the exact loss trajectories and report the average loss by cluster (9i). For each cluster, we provide a label based on the top POS tags of tokens in the cluster and the top 10 contexts closest to the cluster medoid. We report the 3 contexts closest to the cluster medoid. Clustering on the loss trajectories only discovers a relatively simple skill, continuing nouns composed of multiple tokens. POLCA breakthrough clustering recovers a similar skill in Figures 10i and 10ii as well as discovering other skills.

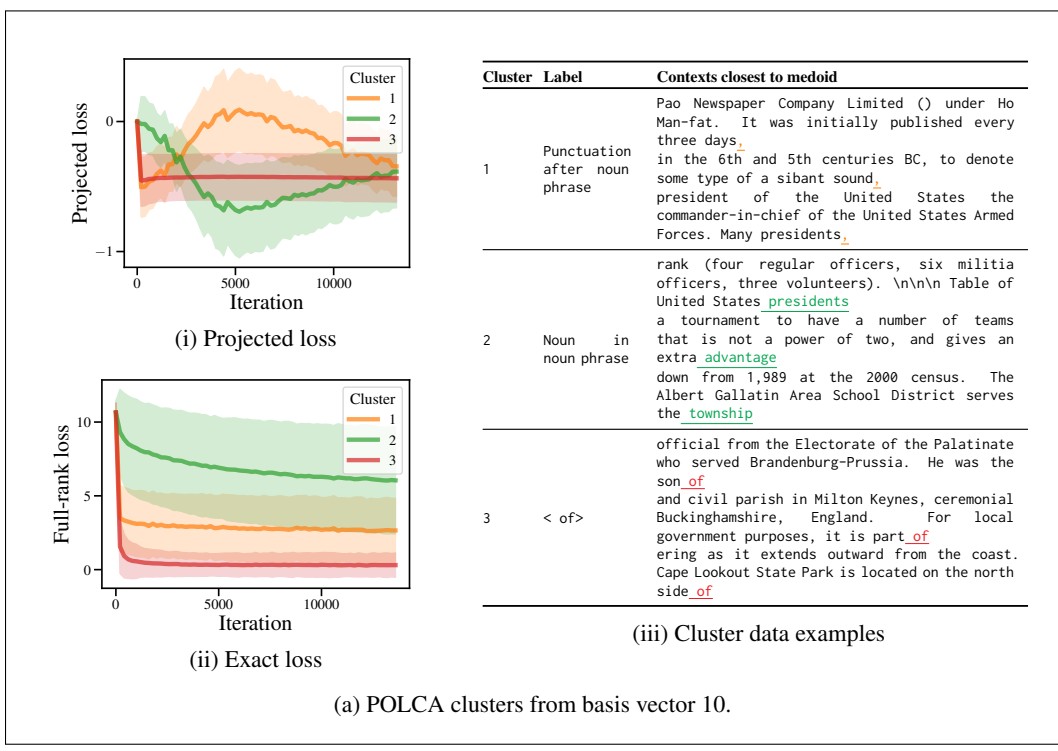

(a) POLCA clusters from basis vector 10.

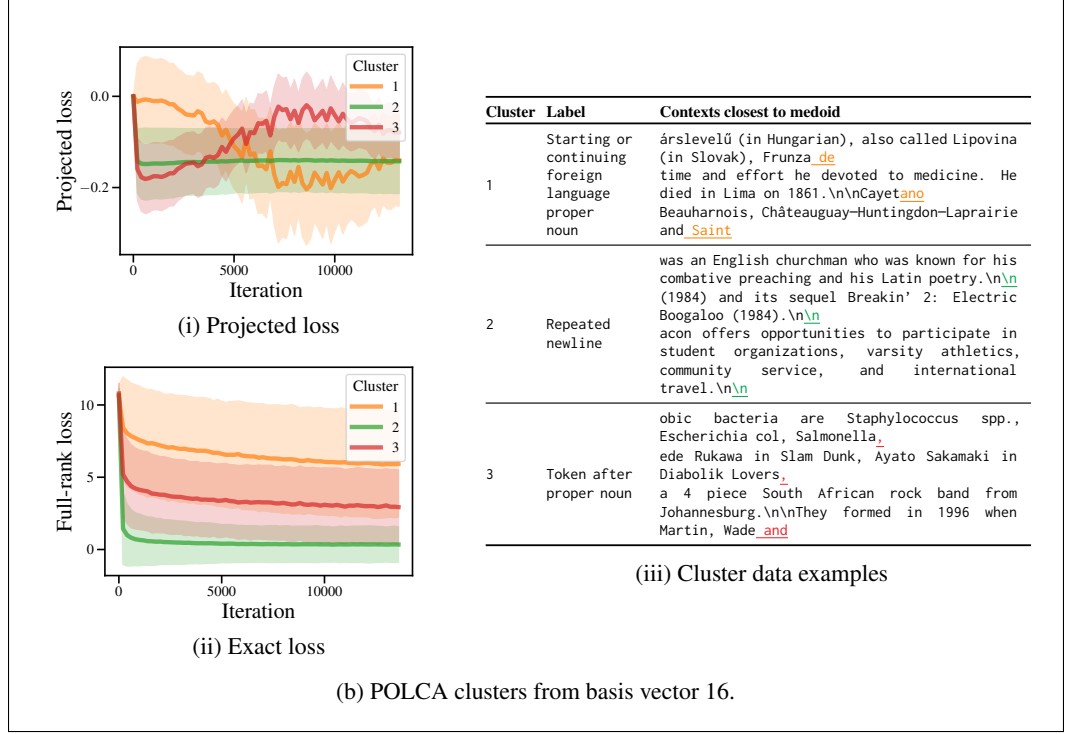

(b) POLCA clusters from basis vector 16.

Figure 10: **English language modeling data clusters with POLCA.** We compute breakthrough clustering on POLCA trajectories for each vector and report the average decomposed POLCA trajectories (10ai and 10bi). Figures 10aii and 10bii show the average of the per-token loss trajectories for each of the clusters found using the POLCA trajectories. For each cluster, we provide a label based on the top tokens in the cluster and the top 10 contexts closest to its medoid. We then report the 3 contexts closest to the cluster medoid. Clustering on the decomposed POLCA trajectories reveals breakthroughs at points in training where the per-token loss curve remains smooth.

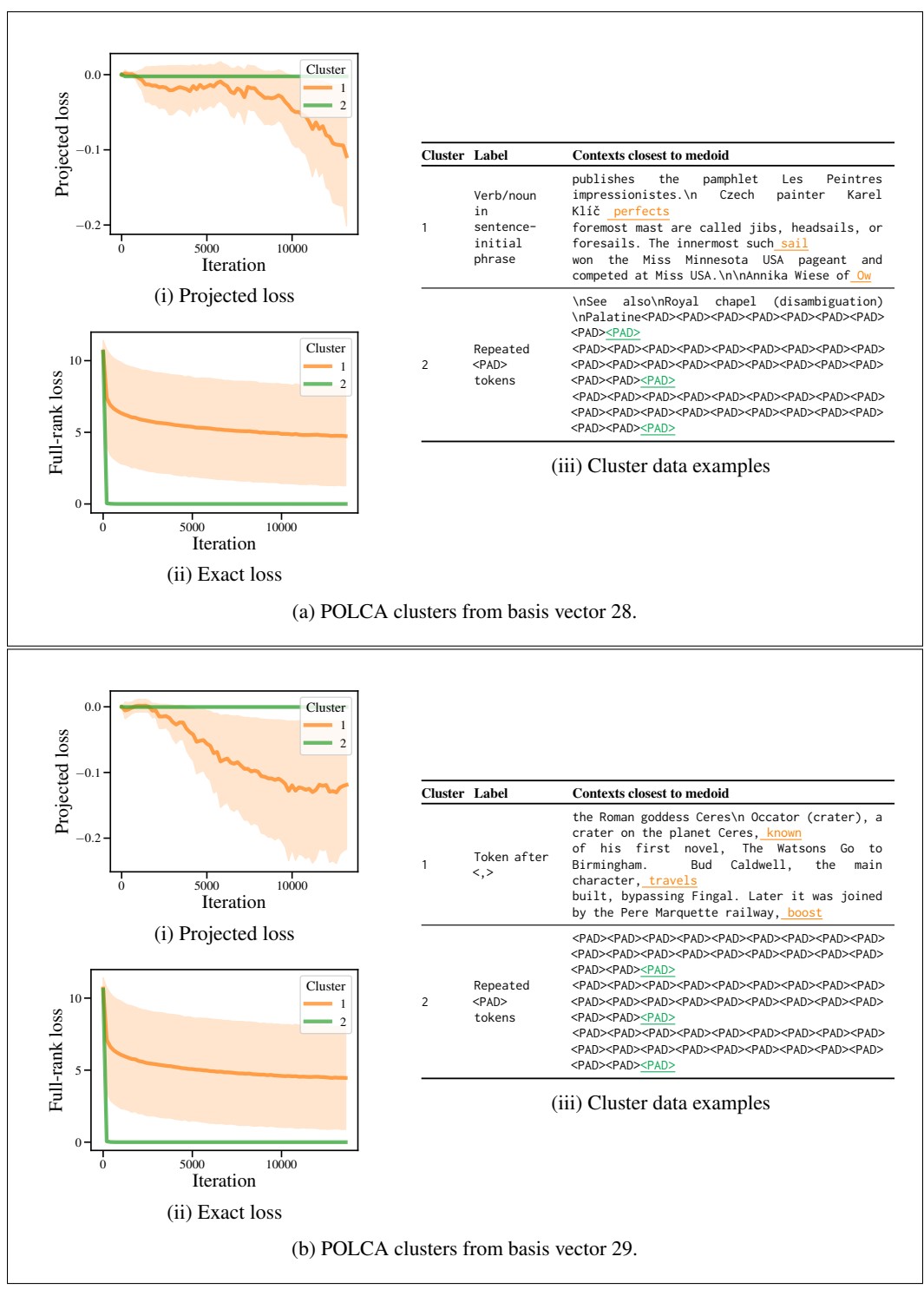

Figure 11: **English language modeling data clusters with POLCA.** We compute breakthrough clustering on POLCA trajectories for each vector and report the average decomposed POLCA trajectories (11ai and 11bi). Figures 11aii and 11bii show the average of the per-token loss trajectories for each of the clusters found using the POLCA trajectories. For each cluster, we provide a label based on the top tokens in the cluster and the top 10 contexts closest to its medoid. We then report the 3 contexts closest to the cluster medoid. Clustering on the decomposed POLCA trajectories reveals breakthroughs at points in training where the per-token loss curve remains smooth.

