# OpenReview forum: "Hidden Breakthroughs in Language Model Training"
_ICLR.cc/2026/Conference — ICLR 2026 Poster_

### Official Review · Reviewer_sGxR · 2025-10-28

**Soundness:** 3
**Presentation:** 2
**Contribution:** 3
**Rating:** 6
**Confidence:** 4

**Summary:**

The authors introduce a new method (POLCA) for decomposing per-sample loss trajectories over training and validate this clustering as an interpretability technique in a synthetic arithmetic setting and a natural language setting.

The methodology involves approximating the change in per-sample loss between successive checkpoints using a second-order Taylor approximation in the weight update restricted to a subspace constructed from the top-k eigenvalues of the Hessian at each of a set of $T$ intermediate checkpoints. For efficiency, the authors additionally approximate the aggregate Hessian using a per-sample Hessian. This decomposes the loss into a set of "projected losses," one for each basis element.

To use POLCA for interpretability, the authors cluster the projected loss trajectories. They demonstrate that clustering the projected losses splits samples into interpretable clusters and provide evidence that this beats baselines like clustering (non-decomposed) per-sample losses.

**Strengths:**

**Problem**: The problem of finding ways to decompose the training trajectory is important and of growing interest, given the rise of the communities working on learning dynamics and developmental interpretability.

**Technique**: The general approach of modeling updates to losses using a Taylor approximation (plus various additional approximations) and clustering trajectories seems sensible and clever. Some of the particular choices around basis construction seem rather ad hoc. Still, this technique seems likely to become a go-to method and baseline for decomposing loss trajectories.

**Weaknesses:**

**Interpretation**: Section 4 and the Figure 3–4 captions are missing a discussion of what "recover the skill of X" means. My concern here would be largely addressed if there were a description of how the ground-truth is established in the arithmetic setting (how do you know what the "skills" are here?). Section 5 is in a better place because it includes a discussion of how clusters are automatically labeled, but lines 450–451 are still vague. Does this mean that you manually come up with names for the labels in Figures 5 and 9–11? If this involves a manual labeling stage, then the tables in these figures should also report how accurate these hypotheses are (i.e., what fraction of in-cluster samples are explained by the given hypothesis?) It should be relatively simple to write automated tests once the tests have been formulated.

If you address all of the above concerns, then I will change my review from a "weak accept" to an "accept." Note: I am open to being convinced to change this even if you do not manage to run substantive new experiments and focus on making the writing around this clearer.

**Benchmarks**: The baselines and ablations in appendices G and H are solid, but the benchmarking is still quite weak. In particular, I don't understand why I should trust the "carry skill homogeneity" metric. I would like to see some benchmarking on additional metrics/tests in the natural language setting. I'm sure it's possible to derive suitable benchmarking metrics from the automatic labeler.

If you address my previous concerns and significantly buttress the benchmarking, then I will change my review to a "strong accept."

**Compute**: The paper is missing a discussion of computational complexity. Hessians are expensive, even when you restrict to the top-k eigenvalues. This could be in the appendices. (The rest of this paragraph is optional.) It would also be valuable to include a (brief) comparison of the computational complexity against the baselines. In particular, I'm interested in understanding how much more compute-intensive second-order POLCA is versus first-order POLCA. If you get second-order POLCA essentially for free, then that would be worth mentioning as a strength in the main body.

**Hyperparameters**: It is not clear what hyperparameters are being used for POLCA. This would be addressed by an additional set of tables in the appendices detailing the hyperparameters used for the Hessian estimation and the hyperparameters used for HDBSCAN. The paper would also benefit from some additional ablations to POLCA hyperparameters. I'd also be interested in a comparison to other clustering algorithms (though this is a marginal addition).

**Questions:**

Questions:
- Is $B$ in (2) the same as $B_T$ from Section 3? Or is it $B_t$ for each individual timestep?
- Some of these cluster error bars are super wide in figures 10–11. That suggests to me that the clustering technique is running into problems. Why cluster just the projected losses and not the full $Tk$ decomposed loss vector? What other ways have you thought about for constructing the basis? Why this one?
- Do you really need this evolving Hessian? Why not just project against (the top Tk eigenvectors of) the final Hessian?
- How accurate is this aggregate-to-per-sample-Hessian approximation in Appendix C really? Do you have an empirical comparison?

Suggestions:

**Improve the description of the method.** I found difficult to parse lines 224–240 the first time around. I didn't realize that "we use a second-order approximation" here referred to "[adding on an additional quadratic term in (1)]" rather than "we use a second-order approximation [in order to choose the basis B]."

My recommendation:
- In (1), show the full second-order expansion.
- Subsequently, define the LCA as the first-order term in this expansion. This would be a new equation (2)
- On line 224, rephrase to "we use the full second-order approximation in (1)"
- On line 235, rephrase to "Exact computation of the second-order term in (1) would be intractable."

**Put the clusters online!** It would be super valuable to be able to browse through the clusters on an interactive website. LLMs make this super easy nowadays.

---

> ### Author Response · Authors · 2025-11-21
> **Response to Reviewer sGxR**
>
> We thank the reviewer for their feedback and positive comments on how the problem is “important and of growing interest” with an approach that “seems sensible and clever” and is “likely to become a go-to method and baseline for decomposing loss trajectories.”
>
> **Response to Weaknesses:**
>
> > Interpretation
>
> We agree that the description here was unclear and have revised Sections 4 and 5 and Appendix E to include more clear descriptions of how exactly we label the ground truth skills. In Section 4, we automatically label the tokens with ground truth skills with functions that use the definition of the “digit” and “carry” skill. We show an example of this labeling in Figure 2. In this example, the input to the model is “907+526=” and the ground truth tokens are “1433”. Here, the tokens are labeled as:
> - 1: (1000s, carry)
> - 4: (100s, no carry)
> - 3: (10s, carry)
> - 3: (1s, no carry)
>
> We label the digit skill based on the position of the token in the output: the first token is 1000s, the second token is 100s, the third is 10s, and the fourth is 1s. We label the carry skill by computing the sum of the next lowest digit place and determining whether it resulted in a carry to the current token. In the example above, we label the 10s place with “carry” because the two numbers in the 1s place (7 + 6) add up to 13, so the 1 is carried to the 10s place, and the 10s place is computed by adding 0 + 2 + 1 = 3, not just 0 + 2 = 2. We have updated Appendix E.1 to describe the labeling more precisely.
>
> In Section 5, we label each token and the three tokens before it using spacy for part-of-speech (POS) tagging. This produces a sequence of four POS tags as the label for each token. To label the clusters, we compute the frequency of each POS tag at each index in the sequences. For each index in the sequence, we then automatically label the cluster with the smallest set of POS tags at that index required to make up 70% of the cluster. For example, if we call the token instance being clustered *t*, one example labeling is the following:
> - 3 tokens before *t*: DET 0.3, NOUN 0.2, PUNCT 0.2
> - 2 tokens before *t*: NOUN 0.2, ADJ 0.2, DET 0.2, PROPN 0.2
> - 1 token before *t*: NOUN 0.5, PROPN 0.3
> - *t*: PUNCT 0.75
>
> We then filter out any labels that require more than 2 POS tags to describe the cluster. In this instance, this would leave: 1 token before *t* (NOUN and PROPN), and *t* (PUNCT). The automatic label for this cluster would then be PUNCT 1 token after NOUN and PROPN. To generate the labels reported in Figure 5, we manually refine the automatically generated label by looking at the top 10 contexts closest to the medoid of the cluster. While these manually refined labels are challenging to verify automatically, we ensure that the contexts closest to the medoid follow the assigned label and that the manually refined label follows the automatically generated label.
>
> > Benchmarks
>
> We have provided recall and F1 score (since homogeneity is equivalent to cluster-wise precision or purity) in Appendices G and H of the revised paper. We also provide comparisons of the fraction of clusters with hidden breakthroughs in the arithmetic setting in Table 1 and Appendices G and H of the revised paper. These new results corroborate our existing results showing that clustering on POLCA trajectories provides higher quality clusters with respect to complex skills than clustering on the exact loss.
>
> > Compute
>
> The two main challenges with scaling the approach are the Hessian basis computation and the frequency of checkpoints. For larger models, computing the Hessian is quite expensive; however, we show in Appendix H that it is likely possible to use a different basis with limited impact on the cluster quality. In terms of the checkpoint frequency, the small scale of the models that we use in our experiments allows for very high granularity of checkpoints used to compute both the basis and the POLCA trajectories. We note that for larger models, it may be very computationally expensive to sample at such high frequency. We have added additional discussion of the computational challenges of scaling this approach to the limitations section of the revised paper.
>
> If the Hessian basis is used for POLCA, then computing second-order POLCA requires very limited additional compute as compared to first-order POLCA. The only additional steps required are to keep track of the eigenvalues when computing the Hessian basis and to track the total gradient at each checkpoint. Then, we can compute the additional second-order term by using these stored values. We have added this comment to Section 3.2.
>
> > Hyperparameters
>
> We have added additional tables in Appendix D describing the hyperparameters used for POLCA and clustering.

---

> > ### Author Response · Authors · 2025-11-21
> > **Response to Reviewer sGxR Questions**
> >
> > **Response to Questions:**
> >
> > > Is B in (2) the same as BT from Section 3? Or is it Bt for each individual timestep?
> >
> > Thank you for pointing this out. $B$ here should be $B_T$ from Section 3. We have fixed this typo in the revised draft.
> >
> > > Some of these cluster error bars are super wide in figures 10–11. That suggests to me that the clustering technique is running into problems. Why cluster just the projected losses and not the full Tk decomposed loss vector? What other ways have you thought about for constructing the basis? Why this one?
> >
> > We note that due to the nature of HDBSCAN clustering, the clusters can have low density but similar trends, which is the main reason for the wide error bars. We are mostly interested in clusters that have similar shaped trajectories rather than the magnitude of the change at each timestep, since they follow similar patterns in the projected loss.
> >
> > We cluster on the projected losses because each token can correspond to multiple skills. For instance, in the arithmetic setting, each token is labeled with both the carrying and digit skill. If we cluster on the entire decomposed loss vector, we lose this ability to distinguish between multiple skills that are learned for the same token. We agree that our current choice of Hessian basis is not necessarily optimal. We have tested a variety of additional basis constructions, such as a stacked Jacobian, Hessian computed using a sliding window, and Hessian computed at the end of training, and chose this construction since it had the best empirical results in the synthetic setting. However, our approach can be used with an arbitrary choice of basis depending on the intended use case for the clusters. Some other choice of basis might represent concepts more cleanly. We have updated Section 3 and Appendix H to describe this more clearly.
> >
> > > Do you really need this evolving Hessian? Why not just project against (the top Tk eigenvectors of) the final Hessian?
> >
> > We use the evolving Hessian to capture directions that are important during the course of training and not just in the final model. We note that this produces some directions that represent short-term oscillation rather than long-term process and filter these directions out by ignoring basis vectors for which the average projected loss increases. However, we want to keep some notion of locally important directions in order to capture breakthroughs in early or intermediate stages of training that may not necessarily be along top directions in the final Hessian. We have updated Section 3.1 to include this discussion.
> >
> > > How accurate is this aggregate-to-per-sample-Hessian approximation in Appendix C really? Do you have an empirical comparison?
> >
> > Thank you for providing this suggestion. We will update the revised paper with the results of this experiment in the next few days.
> >
> > **Response to Suggestions:**
> >
> > > Improve the description of the method. I found difficult to parse lines 224–240 the first time around. I didn't realize that "we use a second-order approximation" here referred to "[adding on an additional quadratic term in (1)]" rather than "we use a second-order approximation [in order to choose the basis B]."
> >
> > We want to clarify that equation (1) describes LCA, equation (2) describes the first-order POLCA decomposition, and then we explain the second-order decomposition in lines 224-240 before defining the full POLCA decomposition in equation (7). Our POLCA approach is significantly different from second-order LCA, since we allow for working with an arbitrary basis rather than parameter-wise decomposition. We have revised line 225 in Section 3 to clarify this distinction.
> >
> > > Put the clusters online! It would be super valuable to be able to browse through the clusters on an interactive website. LLMs make this super easy nowadays.
> >
> > Thank you for the suggestion. We will look into making a convenient website for displaying the clusters and make sure to provide code to reproduce our experiments.

---

> > > ### Author Response · Authors · 2025-11-27
> > > **Result for accuracy of approximation in Appendix C**
> > >
> > > We have *added an additional experiment* in Appendix C of the revised paper showing that empirically (for a subsample of tokens, POLCA checkpoints, and basis vectors), there is an RMSE of 0.145 between the ground truth and approximate second-order terms, indicating that our approximation closely follows the ground truth second-order term.
> > >
> > > We hope our responses have addressed the points you raised and look forward to any additional feedback that you may have.
> > >
> > > Thank you for your review and comments.

---

### Official Review · Reviewer_VeXn · 2025-10-31

**Soundness:** 3
**Presentation:** 3
**Contribution:** 3
**Rating:** 6
**Confidence:** 4

**Summary:**

The manuscript "Hidden breakthroughs in Language Model Training" introduces POLCA, an unsupervised learning analysis of the loss-function trajectory in training. The overall loss is projected onto a suitably defined orthogonal basis and decomposed in contributions for each data point. Subsequently, the decomposed trajectories are analyzed with the HDBSCAN (a density-based and hierarchical) clustering algorithm, chosen mainly for its ability to single out outliers and deal with variable density in the representation space.

**Strengths:**

The authors compare their approach to the previously introduced Loss Change Allocation (LCA) and convincingly debate that the introduced POLCA differs significantly from LCA. The mathematical justification and description is sound and the authors take great care to describe computationally feasible implementation solutions.
As for any unsupervised learning analysis, it is very difficult to define objective performance indicators as well as design demonstrative datasets where the algorithm finds what it is expected to find, without being too trivial. Keeping in mind these inherent difficulties, it is my opinion that the authors do a remarkable job in both aspects.
As seminal idea, the paper is polished enough to be publishable.

**Weaknesses:**

However, I see a few weaknesses:
- The authors use frequently terms like "(conceptual) breakthrough", "phase transition", "skill", without providing a proper scientific definition of them. As much as these terms are used in the related (and cited) literature, it would be better to attempt a self-contained definition to avoid talking to a very specific audience only.
- it looks to me that nothing in the introduced methodology is specific to language models, but rather it could be applied to any (deep) network, with obviously different narrative related to the found clusters. It would be good to comment on this observation.
- It is unclear if there is a somewhat optimal choice for k (for the top k ranked eigenvectors) and the frequency of checkpoints. There is a tension between computational burden and explicative power, but it would be nice to know if there is any suggested "light" analysis one may do preliminarily on the model and training data to assess good values of those parameters.
- I did not find a description on how to, in practice, "discard the oscillatory directions which do not provide an overall decrease in  loss over the course of training according to POLCA." (lines 200-201)

**Questions:**

I'd like the weaknesses points to be addressed

---

> ### Author Response · Authors · 2025-11-21
> **Response to Reviewer VeXn**
>
> We thank the reviewer for their comments and appreciate their positive feedback on how our paper is a “seminal idea” with a “sound” mathematical justification and “[does] a remarkable job” in unsupervised learning analysis.
>
> **Response to Weaknesses:**
>
> > The authors use frequently terms like "(conceptual) breakthrough", "phase transition", "skill", without providing a proper scientific definition of them. As much as these terms are used in the related (and cited) literature, it would be better to attempt a self-contained definition to avoid talking to a very specific audience only.
>
> We thank the reviewer for this suggestion and have revised Section 3.3 and added Section 3.4 to clarify these definitions. We use “breakthrough” and “phase transition” interchangeably to mean a sudden drop in the exact or projected loss and define a “skill” as a specific capability needed for the model to perform a given task.
>
> > It looks to me that nothing in the introduced methodology is specific to language models, but rather it could be applied to any (deep) network, with obviously different narrative related to the found clusters. It would be good to comment on this observation.
>
> We agree that this approach could generalize to other types of deep neural networks. Specifically, it would be interesting to see what types of skills emerge in vision or multimodal models and this is an exciting avenue of future work. We have updated the limitations and future work section of the paper to include discussion of extensions to other types of models.
>
> > It is unclear if there is a somewhat optimal choice for k (for the top k ranked eigenvectors) and the frequency of checkpoints. There is a tension between computational burden and explicative power, but it would be nice to know if there is any suggested "light" analysis one may do preliminarily on the model and training data to assess good values of those parameters.
>
> The optimal choice for k and the frequency of checkpoints depend on the expected dimensionality of the important directions in the model. In more simple settings (such as the synthetic task), the main skills can be recovered with only a few basis vectors, whereas more complex settings (like causal language modeling) have unique skills across a high number of basis directions.  Ideally, the choice of k and granularity should depend on these factors as well as computational constraints. In a setting with complex or noisy data, the frequency should be maximized (within computational constraints). Then, k can be increased if local directions are more important for a specific task. We note that these hyperparameter choices essentially select how to compress the model training trajectory, and similarly to choosing the dictionary size for sparse autoencoders [1], it will likely take additional empirical analysis in a variety of different settings to establish general best practices for these compression decisions. We have revised the paper to include discussion of hyperparameter choices in Appendix D.
>
> > I did not find a description on how to, in practice, "discard the oscillatory directions which do not provide an overall decrease in loss over the course of training according to POLCA." (lines 200-201)
>
> We discard the oscillatory directions by removing any basis vectors for which the projected loss increases on average (ie the final projected loss is higher than the initial projected loss). We choose to do so because an increase in projected loss indicates that the model did not learn along this specific direction. We have clarified this point in Section 3.1 of the revised paper.
>
> [1] Gao et. al. Scaling and evaluating sparse autoencoders, 2024.

---

> > ### Author Response · Authors · 2025-11-27
> >
> > We hope our responses have resolved the weaknesses you raised and would appreciate any other comments or suggestions that you may have.
> >
> > Thank you for your feedback and time invested in the review.

---

### Official Review · Reviewer_rAUf · 2025-10-31

**Soundness:** 2
**Presentation:** 4
**Contribution:** 3
**Rating:** 4
**Confidence:** 3

**Summary:**

This work investigates how training breakthroughs, sharp decreases in training loss attributable to a meaningful change in the model’s predictions, can be automatically discovered. Analyzing the loss aggregated over all examples can allow for the discovery of some breakthroughs. However, the authors argue that the interplay of how the model learns new skills, as well as how different examples may require different skills for correct prediction, leaves some breakthroughs obscured. To discover new breakthroughs, the authors propose a new approach, POLCA, that determines using a Taylor approximation the change in loss attributable to a specific direction in parameter space. Applying clustering methods to the projected loss of specific examples helps elucidate human interpretable features that the model has learned in order to decrease the loss. The authors experiment with their approach on a synthetic addition task with pre-specified skills, as well as a general task of language modelling over a natural language dataset. They demonstrate that POLCA can discover breakthroughs that are obscured in the aggregated, unprojected loss.

**Strengths:**

- The paper is well presented, with clear writing, high quality and intuitive figures, and a strong argumentative flow.
- The selected problem of discovering training breakthroughs is interesting and pertinent to many areas, such as interpretability and the study of model training dynamics.
- The new method, POLCA, is intuitive, well justified theoretically, and provides a satisfying means for attributing changes in loss to different directions in the parameter space. While it bears a strong similarity to its predecessor LCA, the change to an arbitrary basis over the parameter space is important, as these training breakthroughs should not be expected to be attributable to individual weights in the model.
- The chosen experimental settings (synthetic counting task and natural language modelling) are good choices for both explanatory and argumentative purposes.
- Some presented results are very interesting and convincing. For instance, demonstrating that there is a clear clustering on digit but not on carry skill (Figure 3) is interesting, and justifies the need for additional methods for discovering training breakthroughs.

**Weaknesses:**

- [W1] The results presented are not entirely convincing in terms of showing that POLCA discovers training breakthroughs based on human interpretable features. In figure 4c and section 4.2, for instance, the authors claim that the first basis vector “recovers the digit skill”. However, clusters 1 and 3 are composites of multiple different digit positions, so it is difficult to say that all of those examples are having their loss improve because of the same skill, especially since one digit is excluded.
- [W2] It is not always clear that the projected loss trajectories imply a training breakthrough. For figure 4a, does the slight decline of cluster 1 imply a training breakthrough? What is the implication that cluster 3 has a sharper projected loss drop, but has very few examples prescribed to it? Figure 5 shows directions where projected loss for some examples increases while for others it decreases. Why would the understanding of one human interpretable concept reduce understanding of a related concept?
- [W3] It’s unclear how the method should be applied given certain methodological choices and the results presented. See [Q1] and [Q2].

**Questions:**

- [Q1] Notably only two of the 22 basis vectors collected for the experiment in Section 5.1 are presented in the main text, and 5 in the Appendix. This is after discarding several basis vectors as well. How should one think of these vectors, meant to represent a “breakthrough” since they present a mean decrease in the loss, when they also don’t provide a meaningful interpretation? Is POLCA meant to provide a set of results that have to be further interpreted and classified, rather than presenting a set of vectors that are inherently important?
- [Q2] On line 315 and 316, it is mentioned that tokens are discarded if their loss increases along a given basis vector. How many tokens are typically discarded this way? If some tokens have their loss increase along this direction, how can we know it still represents a training breakthrough?
- [Q3] Line 198 of section 3.1 states that by repeatedly adding top eigenvectors from successive checkpoint Hessians, directions of long-term stable movement will be added. However, since each successive Hessian is projected onto the nullspace of the current set of vectors, is this not more likely to collect directions reflective of local oscillations, as the stable directions are likely already represented in the collected set of vectors?

---

> ### Author Response · Authors · 2025-11-21
> **Response to Reviewer rAUf**
>
> We thank the reviewer for their comments and their positive feedback on how the problem is “interesting and pertinent to many areas”, our POLCA method is “intuitive, well justified theoretically, and provides a satisfying means for attributing changes in loss to different directions in the parameter space”, and our paper is “well presented” with “clear writing” and “high quality and intuitive” figures.
>
> **Response to Weaknesses:**
>
> > [W1] The results presented are not entirely convincing in terms of showing that POLCA discovers training breakthroughs based on human interpretable features. In figure 4c and section 4.2, for instance, the authors claim that the first basis vector “recovers the digit skill”. However, clusters 1 and 3 are composites of multiple different digit positions, so it is difficult to say that all of those examples are having their loss improve because of the same skill, especially since one digit is excluded.
>
> We note that in our analysis in Figure 4, we restrict the minimum HDBSCAN cluster size to be large (> 150) in order to ensure that the clusters contain a significant number of tokens. This selection results in 2-3 clusters per basis vector, which will inherently cause some grouping of digits together. To clarify, our results for digit clustering on POLCA show that we recover the 1000s place skill in the first dimension (later vectors recover other digit skills). We have revised the writing in Section 4.2 to clarify this point.
>
> > [W2] It is not always clear that the projected loss trajectories imply a training breakthrough. For figure 4a, does the slight decline of cluster 1 imply a training breakthrough? What is the implication that cluster 3 has a sharper projected loss drop, but has very few examples prescribed to it? Figure 5 shows directions where projected loss for some examples increases while for others it decreases. Why would the understanding of one human interpretable concept reduce understanding of a related concept?
>
> A training breakthrough in this context is a point where the projected or exact loss for some subset of data decreases via sudden acceleration during training. We have added Section 3.4 to provide precise definitions of breakthroughs and hidden breakthroughs and have added results in Table 1 showing that POLCA produces the highest fraction of clusters with hidden breakthroughs as compared to other decomposition strategies. We note that a smaller magnitude of projected loss for a cluster (e.g. cluster 1 in Figure 4a) still represents a breakthrough, just one of smaller magnitude than that in the other cluster. Furthermore, clusters with conflicting trends still can represent breakthroughs. This is because models must use many different capabilities in order to solve a given task. In the synthetic setting (Fig 4), the task by design is composed of very few skills that the model must learn, so there is less opportunity for conflict between different capabilities. On the other hand, in the natural language setting, many skills seemingly conflict with each other. For instance, Clusters 1 and 3 in Figure 5(a)(i) represent clusters that could conceivably conflict, as they represent ending different types of phrases with < from> or < to> (Cluster 1) versus < ,> (Cluster 3). The model must learn both of these skills for different phrases, but because they produce different outputs, learning one may prevent the output required for the other skill. We have updated our discussion in Sections 4.2 and 5.2 of the revised paper to clarify breakthroughs in the case of conflicting concepts.
>
> > [W3] It’s unclear how the method should be applied given certain methodological choices and the results presented. See [Q1] and [Q2].
>
> Please see the responses to [Q1] and [Q2].

---

> > ### Author Response · Authors · 2025-11-21
> > **Response to Reviewer rAUf Questions**
> >
> > **Response to Questions:**
> > > [Q1] Notably only two of the 22 basis vectors collected for the experiment in Section 5.1 are presented in the main text, and 5 in the Appendix. This is after discarding several basis vectors as well. How should one think of these vectors, meant to represent a “breakthrough” since they present a mean decrease in the loss, when they also don’t provide a meaningful interpretation? Is POLCA meant to provide a set of results that have to be further interpreted and classified, rather than presenting a set of vectors that are inherently important?
> >
> > We clarify that we collect 30 basis vectors total (26 after removing directions with an increase in loss) and our labeling approach finds homogeneous clusters for 22 of these directions. In the paper, we show a selection of directions with interesting clusters and trends in the projected loss. However, 15 out of the 23 directions not shown still contain homogeneous clusters and have an overall decrease in loss.
> >
> > Furthermore, as we state in footnote 1 in Section 5, our automated labeling approach does not capture all interpretable concepts. It is limited to lexical and syntactic skills. This labeling strategy allows for unsupervised, automatic identification of these syntactic skills and ensures strict interpretable labels. However, it fails to capture many human-interpretable language modeling skills. The discarded vectors may (and likely do) contain other interpretable skill clusters that are not found by automated labeling. We have revised Section 5.1 and the limitations section to clarify this interpretation.
> >
> > > [Q2] On line 315 and 316, it is mentioned that tokens are discarded if their loss increases along a given basis vector. How many tokens are typically discarded this way? If some tokens have their loss increase along this direction, how can we know it still represents a training breakthrough?
> >
> > We discard an average of 2360.8 out of 5000 arithmetic tokens and 6655.5 out of 12600 natural language tokens per direction in this way and have revised the paper to include this information. We note that breakthroughs happen for subsets of tokens, not every token, along a given basis vector. These subsets are associated with specific skills that improve along that direction. If an example increases its loss along that direction, we would not claim the direction corresponds to a breakthrough in predicting that example. Notably, all vectors that we consider have a mean decrease in projected loss, so they still represent directions of long-term learning and not just oscillation. We have included this clarification in Sections 4.1 and 5.1 of the revised paper.
> >
> > > [Q3] Line 198 of section 3.1 states that by repeatedly adding top eigenvectors from successive checkpoint Hessians, directions of long-term stable movement will be added. However, since each successive Hessian is projected onto the nullspace of the current set of vectors, is this not more likely to collect directions reflective of local oscillations, as the stable directions are likely already represented in the collected set of vectors?
> >
> > We agree that our current choice of Hessian basis is not necessarily optimal. To account for the limitation you describe, we require an extra step that discards directions where loss does not decrease. However, our approach can be used with an arbitrary choice of basis. We hope future work will explore principled bases that favor long-term movement by construction. We have updated Section 3 to describe this more clearly.

---

> > > ### Author Response · Authors · 2025-11-27
> > >
> > > We hope our responses have resolved your concerns and look forward to any other feedback that you may have.
> > >
> > > Thank you for your time and consideration.

---

### Official Review · Reviewer_f3uN · 2025-11-01

**Soundness:** 3
**Presentation:** 4
**Contribution:** 3
**Rating:** 6
**Confidence:** 4

**Summary:**

The authors propose to decompose the learning process of neural networks into contributions based on subsets of the data and directions in parameter space, so as to more clearly exhibit “hidden” phase transitions or breakthroughs corresponding to individual concepts. This kind of study of neural network development is an emerging area of interest and they propose a new simple and (perhaps) scalable methodology POLCA for discovering these breakthroughs and attributing them to patterns in the data distribution. This methodology is tested in two settings: multi-digit addition where interpretable breakthroughs are found by the method, and language model training (40m parameters) where the results are interpretable but more mixed (in my opinion).

**Strengths:**

* Well motivated problem, and a clean derivation of a simple methodology for addressing it
* The paper is very clearly written, with appropriate and well-captioned figures
* Some quite compelling results in a toy setting with addition and carrying
* The automatic labelling for clusters in the language model setting seems to be well-done, I found this interesting in its own right.

**Weaknesses:**

* Major: I am not completely convinced by the framing of “breakthroughs in the loss” being discovered in the smooth learning curve via POLCA for the larger language models. If I can summarise (8) we decompose the loss into a sum of components, which may be positive or negative. By their nature (since they depend on narrower subsets of the data distribution and directions in parameter space) these components will tend to have much more variety in their behaviour over training, and by construction them sum to something we know is generically fairly featureless. It is therefore no surprise to see empirically that there is such variation. This seems not sufficient in my mind to justify language like “breakthroughs in the loss projected onto that basis” in e.g. Fig 5\. In both cases the projected losses either seem fairly uninteresting (cluster 2 in 5(a)(i) and cluster 3 in 5(b)(i)) or non-monotonic. Could the authors elaborate further on why I should look at these plots and see a strong analogy to Fig 4 (where I do agree with the interpretation of hidden breakthroughs).
* Minor: in both spirit and methodology the study here seems quite close to that used by Michaud et al (2024) and I think it is worth addressing the novelty of POLCA relative to that work directly.
* Minor: I think many of the figures with training steps on the x-axis would benefit from being plotted in log-scale. For instance all the interesting information in Fig 4 is compressed into the very beginning of the plot, making it hard to distinguish the clusters.

**Questions:**

* What is the reasoning for excluding basis vectors where the loss increases? It does seem like even when projected onto the remaining basis vectors loss increases are typical for many clusters at some point in training.
* The only limit to scalability I see here is approximation of Hessians, which is quite memory intensive at scale. The experiments done here are in relatively small models but I do not doubt this could be done at larger scale. Could the authors comment on any significant limitations here?
* The aggregated loss decreases, but it has been widely observed that some individual token losses have non-monotonic behaviour. Even if we see a clear “breakthrough” trend where the projected loss plateaus, decreases and then plateaus, I’m unsure whether I would think of that as a genuine breakthrough. It would seem to depend on how semantically coherent the cluster is. Can the authors confirm that this concern is what is motivating the details in the “automatic labelling” section e.g. the 70% threshold? I may be incorrect, but this “unity” of the cluster seems to be quite a crux for the methodology and I’d appreciate seeing more detail on it.

---

> ### Author Response · Authors · 2025-11-21
> **Response to Reviewer f3uN**
>
> We thank the reviewer for their feedback and appreciate their positive comments on the “well-motivated problem”, “clean derivation”, “compelling results”, and “clearly written” paper.
>
> **Response to weaknesses**
>
> > Major: I am not completely convinced by the framing of “breakthroughs in the loss” being discovered in the smooth learning curve via POLCA for the larger language models.
>
> We argue that clusters with non-monotonicity still exhibit breakthroughs. In previous work, it has been extensively documented that in training, models tend to exhibit simplicity bias [1, 2, 3]. That is, early in training, they often learn simple heuristics to solve tasks before learning a more complex solution later in training. This process of learning a simpler but worse solution early in training may cause plateaus and non-monotonicity in the loss [4]. Furthermore, models must use many different capabilities in order to solve a given task. For instance, in the synthetic setting (Fig 4) the model could first learn to always add without carrying, then have a conflict between the carry and no carry datapoints, leading to the non-monotonicity seen early in training in Figure 4(a). In the natural language setting, Clusters 1 and 3 in Figure 5(a)(i) represent clusters that could conceivably conflict, as they represent ending different types of phrases with < from> or < to> (Cluster 1) versus < ,> (Cluster 3). The model must learn both of these skills for different phrases, but learning one may cause a conflict in how the model performs predictions for the other. These clusters thus still represent potential breakthroughs, but represent a case where learning one skill impedes another. We note that the clusters pointed out by the reviewer (Cluster 2 in 5(a)(i) and Cluster 3 in 5(b)(i)) represent simple patterns that are learned quickly by the model. These do not necessarily represent “hidden” breakthroughs, but are still homogeneous clusters that provide contrast to skills that involve breakthroughs.
>
> We have added Section 3.4 to describe the exact definition of breakthroughs and hidden breakthroughs and have *added new results showing how often hidden breakthroughs occur* in Table 1 and Appendices G and H. We have updated our discussion in Sections 4.2 and 5.2 of the revised paper to clarify why breakthroughs can still occur in non-monotonic clusters.
>
> > Minor: in both spirit and methodology the study here seems quite close to that used by Michaud et al (2024) and I think it is worth addressing the novelty of POLCA relative to that work directly.
>
> There are some similarities between our work and that of Michaud et al (2024), but we note that they are interested in scaling, whereas we focus on model training dynamics. Moreover, Michaud et al state that polygenic scaling effects (datapoints that require multiple skills and thus have multiple breakthroughs) are a limitation of their method. POLCA is designed to address this limitation directly, as it decomposes each datapoint’s trajectory along multiple directions. We have revised the paper to include additional discussion of this in Section 2.
>
> > Minor: I think many of the figures with training steps on the x-axis would benefit from being plotted in log-scale.
>
> We thank the reviewer for this comment. We tested plotting in log scale and with a trimmed x axis and found that the trimmed axis best showed the breakthrough periods the clusters, so we have trimmed the x axes in the arithmetic plots in the revised paper.
>
> [1] Nakkiran et. al. SGD on Neural Networks Learns Functions of Increasing Complexity, 2019.
> [2] Shah et. al. The Pitfalls of Simplicity Bias in Neural Networks, 2020.
> [3] Valle-Perez et. al. Deep learning generalizes because the parameter-function map is biased towards simple functions, 2018.
> [4] Rosenfeld et. al. Outliers with Opposing Signals Have an Outsized Effect on Neural Network Optimization, 2023.

---

> > ### Author Response · Authors · 2025-11-21
> > **Response to Reviewer f3uN Questions**
> >
> > **Response to Questions:**
> >
> > > What is the reasoning for excluding basis vectors where the loss increases? It does seem like even when projected onto the remaining basis vectors loss increases are typical for many clusters at some point in training.
> >
> > We exclude basis vectors where the loss increases on average because prior work has found that the directions along which the loss increases represent directions of oscillation (and not learning) during training. The increase in loss indicates that the model has not learned on average along this direction. We have updated Section 4.1 and 5.1 to clarify this point.
> >
> > > The only limit to scalability I see here is approximation of Hessians, which is quite memory intensive at scale. The experiments done here are in relatively small models but I do not doubt this could be done at larger scale. Could the authors comment on any significant limitations here?
> >
> > The two main challenges with scaling the approach are the Hessian basis computation and the frequency of checkpoints. For larger models, computing the Hessian is quite expensive; however, we show in Appendix H that it is likely possible to use a different basis with limited impact on the cluster quality. In terms of the checkpoint frequency, the small scale of the models that we use in our experiments allows for very high granularity of checkpoints used to compute both the basis and the POLCA trajectories. We note that for larger models, it may be very computationally expensive to sample at such high frequency. We have added additional discussion of the computational challenges of scaling this approach to the limitations section of the revised paper.
> >
> > > The aggregated loss decreases, but it has been widely observed that some individual token losses have non-monotonic behaviour. Even if we see a clear “breakthrough” trend where the projected loss plateaus, decreases and then plateaus, I’m unsure whether I would think of that as a genuine breakthrough. It would seem to depend on how semantically coherent the cluster is. Can the authors confirm that this concern is what is motivating the details in the “automatic labelling” section e.g. the 70% threshold? I may be incorrect, but this “unity” of the cluster seems to be quite a crux for the methodology and I’d appreciate seeing more detail on it.
> >
> > Yes, your understanding is correct, we wanted to confirm that the discovered clusters are clean enough that an automatic labeller could identify meaningful patterns, and it did. We are interested in breakthroughs corresponding to a specific human interpretable skill, so we analyze only clusters which are sufficiently homogeneous: those dominated by examples of a specific automatically labelled pattern.  If the cluster contains many points that do not exhibit that skill, their shared trend may result from coincidental alignment, rather than a shared skill. Of course, our automatic labeler searches for very limited pattern templates, so even if it does not identify a simple label, the cluster could correspond to an interpretable skill.

---

> > > ### Author Response · Authors · 2025-11-27
> > >
> > > We hope our responses have addressed your concerns and welcome any other comments or suggestions that you may have.
> > >
> > > Thank you for your review and consideration.

---

### Author Response · Authors · 2025-11-21
**General comment**

We thank the reviewers for their helpful and positive feedback. We appreciate how reviewers thought the method is a “seminal idea” [VeXn] and the problem is “well-motivated” [f3uN], “interesting and pertinent” [rAUf] and “important” [sGxR]. They agreed that our method has a solid theoretical foundation; in particular, reviewer f3uN mentioned the “clean derivation”, while reviewer rAUf said it is “well justified theoretically” and reviewer VeXn noted that the “mathematical justification and description is sound.” The reviewers also found our experiments to have “compelling results” [f3uN] and do a “remarkable job” at unsupervised analysis [VeXn]. We address the main concerns brought up by the reviewers below.

1. **Definition of “breakthroughs in the loss”**

All of the reviewers brought up questions about what exactly a “breakthrough” means in the context of our paper. We have added Section 3.4 to make it clear that a training breakthrough in this context is a set of points that have a sudden change in the projected or exact loss and have run new experiments in Table 1 to compute which clusters have hidden breakthroughs. These experiments verify that POLCA discovers more clusters with breakthroughs than other approaches in the arithmetic setting. We have also revised Section 3, 4, and 5 in the paper to address the specific concerns brought up by each reviewer.

One shared question brought up by reviewers f3uN and rAUf is whether non-monotonic clusters can still represent breakthroughs. We argue that breakthroughs can still occur in non-monotonic clusters because (especially in the more complex natural language setting) there are a variety of skills for which learning one skill can cause changes in how the model performs predictions for the other and thus cause an opposite trend for the other skill. For instance, in the natural language setting, learning to predict a semicolon after an independent clause could affect how the model predicts periods at the end of a sentence.

2. **Clarification of hyperparameter and method choices**

The reviewers ask for further elaboration on our choices for discarding oscillatory vectors [f3uN, rAUf, VeXn] as well as the basis construction [rAUf, sGxR] and automatic labeling [sGxR] and hyperparameter [VeXn, sGxR] details. We have revised the experimental and method sections to clarify these questions and have added details in Appendices D and H to provide a description of the reasoning for our basis construction and hyperparameter choices.

3. **Benchmarking**

Reviewer sGxR requested additional benchmarking metrics in the synthetic and natural language settings. We have added additional standard clustering metrics in Appendices G and H, which further demonstrate that POLCA finds the highest quality clusters with respect to the carrying skill.

4. **Computational complexity**

Reviewers f3uN and sGxR asked for additional discussion of the computational complexity of our method and how it can be applied at scale. The main challenge of scaling is computing the Hessian basis (however, we show in Appendix H that it is likely possible to use a different basis that is easier to compute with limited impact on the cluster quality) and ensuring that checkpoints are sampled at sufficient frequency. We have added additional discussion of the computational challenges of scaling this approach to the limitations section of the revised paper.

---

### Author Response · Authors · 2025-12-03
**Rebuttal summary**

Dear AC,

We believe that we have addressed the concerns brought up by all of the reviewers and summarize how we did so below.

- **Reviewer f3Un:**
    - **Concerns:** framing of “breakthroughs”, non-monotonic tokens, clarifications (comparing to Michaud et. al, excluding basis vectors with loss increase, scalability), presentation (log-scale plots)
    - **Response:** We added Section 3.4 to add a definition of breakthroughs and hidden breakthroughs and ran new experiments in Table 1 and Appendices G and H to compare how often hidden breakthroughs occur for different types of clusters. We addressed the clarifications and presentation issues by revising Sections 2, 4.1, 5.1, and Figures 3 and 4.

- **Reviewer rAUf:**
    - **Concerns:** showing that “breakthroughs” are discovered, non-monotonic breakthroughs, clarifications (discarded tokens, discarded vectors, Hessian basis)
    - **Response:** We wrote Section 3.4 to define breakthroughs, added new experiments in Table 1 and Appendices G and H to quantify how often hidden breakthroughs are found, and revised Sections 4.2 and 5.2 to clarify the reviewer’s points about an unclear definition of skill recovery and the existence of non-monotonic breakthroughs. We revised Sections 3, 4.1, and 5.1 to resolve the clarifications brought up by the reviewer.

- **Reviewer VeXn:**
    - **Concerns:** definition of breakthroughs and skills, clarifications (applications to other types of models, hyperparameters, how to discard oscillatory directions)
    - **Response:** We added Section 3.4 to define breakthroughs and hidden breakthroughs and revised Section 3.3 to define a “skill” in this context. We updated Section 3.1, the limitations section, and Appendix D to address the clarifications requested by the reviewer.

- **Reviewer sGxR:**
	- **Concerns:** interpretation, benchmarking, clarifications (compute, hyperparameters, basis construction, method description)
	- **Response:** We have revised Sections 4 and 5 and Appendix E to address the interpretation concern by discussing how synthetic and natural language skills are labeled. We have provided new metrics (recall, F1 score, hidden breakthrough recovery) in Appendices G and H and Table 1 to add more thorough benchmarking. We have revised Section 3 and Appendices C, D and H to clarify the points brought up by the reviewer.
    - **Note:** Reviewer sGxR stated that if we address the concerns in the “interpretation” section of their review,
>“then I will change my review from a "weak accept" to an "accept." Note: I am open to being convinced to change this even if you do not manage to run substantive new experiments and focus on making the writing around this clearer.”

      We have thoroughly addressed this concern in the revised paper by adding discussion of the interpretation in Sections 4, 5 and Appendix E as well as providing a detailed response to the reviewer.

We sincerely thank the reviewers and AC for their time and constructive comments.

---

### Meta-Review · Area_Chair_VNBC · 2026-01-08

**Summary:**

This paper introduces POLCA, an unsupervised learning analysis of the loss-function trajectory in training, by decomposing per-sample loss trajectories over training. The overall loss is decomposed in contributions for per-sample loss trajectories and then analyzed with the HDBSCAN (a density-based and hierarchical) clustering algorithm, chosen mainly for its ability to single out outliers and deal with variable density in the representation space. It is served as an interpretability technique in a synthetic arithmetic setting and a natural language setting.

Most of the reviewers vote for positive evaluation. After my own reading, this paper is good in general and I suggest the authors to make this paper clear and include the reviewers' feedback. However, there is one acknowledgement section left there, which may avoid the double-blind policy.

**Reviewer Concerns:**

The authors have addressed the reviewers' concern during the rebuttal.

**Reviewer Scores:**

Most reviewers vote for positive evaluation and Reviewer rAUf who gave the negative evaluation may increase the score.

---

### Decision · Program_Chairs · 2026-01-26

Accept (Poster)